# Cristae dynamics is modulated in bioenergetically compromised mitochondria

Mathias Golombek[1] , Thanos Tsigaras[1] , Yulia Schaumkessel[1] , Sebastian Hänsch[2], Stefanie Weidtkamp-Peters[2] , Ruchika Anand[1] , Andreas S Reichert[1] , Arun Kumar Kondadi[1]

Cristae membranes have been recently shown to undergo intra-mitochondrial merging and splitting events. Yet, the metabolic and bioenergetic factors regulating them are unclear. Here, we investigated whether and how cristae morphology and dynamics are dependent on oxidative phosphorylation (OXPHOS) complexes, the mitochondrial membrane potential ($\Delta\Psi_m$), and the ADP/ATP nucleotide translocator. Advanced live-cell STED nanoscopy combined with in-depth quantification were employed to analyse cristae morphology and dynamics after treatment of mammalian cells with rotenone, antimycin A, oligomycin A, and CCCP. This led to formation of enlarged mitochondria along with reduced cristae density but did not impair cristae dynamics. CCCP treatment leading to $\Delta\Psi_m$ abrogation even enhanced cristae dynamics showing its $\Delta\Psi_m$-independent nature. Inhibition of OXPHOS complexes was accompanied by reduced ATP levels but did not affect cristae dynamics. However, inhibition of ADP/ATP exchange led to aberrant cristae morphology and impaired cristae dynamics in a mitochondrial subset. In sum, we provide quantitative data of cristae membrane remodelling under different conditions supporting an important interplay between OXPHOS, metabolite exchange, and cristae membrane dynamics.

## Introduction

Mitochondria are highly dynamic organelles playing vital roles in various cellular functions involving energy conversion, calcium buffering, iron–sulfur cluster biogenesis, immune responses, apoptosis, and various metabolic reactions. Mitochondria consist of a smooth outer membrane (OM) and a heterogenous inner membrane (IM). Furthermore, the IM is spatially divided into the inner boundary membrane (IBM) and cristae membrane (CM). IBM runs parallel to the OM, whereas CM invaginates towards the mitochondrial matrix. Crista junctions (CJs) are located at the interface between IBM and CM. CJs are slot- or pore-like structures typically having a diameter in the range of 25 nm (Frey & Mannella, 2000) which separates the intracristal space and intermembrane space adjacent to the IBM. CJs act as a diffusion barrier for proteins (Gilkerson et al, 2003; Vogel et al, 2006; Wurm & Jakobs, 2006; Davies et al, 2011). In addition, CJs are proposed to play a role in the metabolite diffusion such as ATP, $Ca^{2+}$, cytochrome $c$ (Frey et al, 2002; Mannella et al, 2013). In fact, it has been shown that CJs establish electrochemical boundaries that allow differential membrane potential between the cristae and IBM (Wolf et al, 2019). In addition, individual cristae within a mitochondrion possess disparate membrane potentials demonstrating their functional independency. CJ formation was shown to depend on the high-molecular weight "Mitochondrial Contact Site and Cristae Organizing System" (MICOS) complex consisting of at least seven proteins located at CJs (Rabl et al, 2009; Harner et al, 2011; Hoppins et al, 2011; von der Malsburg et al, 2011). A uniform nomenclature of MICOS subunits was established subsequently (Pfanner et al, 2014). MIC10, MIC13, MIC19, MIC25, MIC26, MIC27, and MIC60 constitute the proteins of the MICOS complex. The loss of CJs is observed upon depletion of most MICOS proteins leading to separation of IBM and CM (Harner et al, 2011; Hoppins et al, 2011; von der Malsburg et al, 2011; Stephan et al, 2020). MIC13 bridges the two subcomplexes: MIC60/19/25 and MIC10/26/27 (Guarani et al, 2015; Anand et al, 2016) via conserved WN and GxxxG MIC13 motifs (Urbach et al, 2021). Some subunits of the MICOS complex are evolutionarily conserved and have an endosymbiotic origin in $\alpha$-proteobacteria (Munoz-Gomez et al, 2015; Eramo et al, 2020; Munoz-Gomez et al, 2023). The MICOS complex proteins are involved in mitochondrial protein import (von der Malsburg et al, 2011), lipid trafficking (Michaud et al, 2016), bending, and remodeling the membranes (Hessenberger et al, 2017; Tarasenko et al, 2017) in addition to their role in the formation of CJs. Mutations of MICOS subunits have been associated with Parkinson's disease, mitochondrial encephalopathy with liver disease and bilateral optic neuropathy, myopathy, and lactic acidosis (Guarani et al, 2016; Tsai et al, 2018; Beninca et al, 2021; Marco-Hernández et al, 2022). Apart from the MICOS complex, $F_1F_O$ ATP synthase and OPA1 (Optic Atrophy Type I) play important roles in cristae remodelling as interplay of

[1]Institute of Biochemistry and Molecular Biology I, Medical Faculty and University Hospital Düsseldorf, Heinrich-Heine-University Düsseldorf, Düsseldorf, Germany
[2]Center for Advanced Imaging, Faculty of Mathematics and Natural Sciences, Heinrich Heine University Düsseldorf, Düsseldorf, Germany

Correspondence: reichert@hhu.de; kondadi@hhu.de

these complexes and cardiolipin is required for formation and maintenance of cristae and CJs (Kondadi et al, 2019, 2020b; Anand et al, 2021).

Cristae exist in various shapes and sizes depending on the cells, tissues, and bioenergetic requirements (Zick et al, 2009). Moreover, alterations in cristae structure have been associated with neuro-degeneration, diabetes, obesity, cardiomyopathy, and myopathies (Zick et al, 2009; Eramo et al, 2020). For several decades, a static view of cristae prevailed based on the early existing electron microscopy (EM) data despite several indications of cristae remodeling. Cells undergoing apoptosis showed a massive reorganization of the mitochondrial IM where it reorganized and interconnected within a short span of few minutes (Scorrano et al, 2002). EM and electron tomography (ET) revealed that when isolated mitochondria were exposed to ADP, the IM confirmation changed to a highly inter-connected network accompanied by matrix condensation termed State III respiration (Hackenbrock, 1966; Mannella et al, 1994; Perkins et al, 1997). Reduction in ADP levels resulted in a drastic decrease of the interconnected cristae network accompanied by matrix expansion or state IV respiration. The application of SR techniques to resolve mitochondrial membranes has conclusively shown that cristae membranes are highly dynamic (Huang et al, 2018; Wang et al, 2019; Kondadi et al, 2020a; Kondadi et al, 2020b; Liu et al, 2022). Recently, we showed using live-cell–stimulated emis-sion depletion (STED) super-resolution (SR) nanoscopy that cristae membranes are dynamic and undergo continuous cycles of mem-brane remodeling dependent on the MICOS complex (Kondadi et al, 2020a). The cristae merging and splitting events are balanced, re-versible, and depend on the presence of the MICOS subunit MIC13. Fluctuation of membrane potential within individual cristae and photoactivation experiments at cristae-resolving resolution support the notion that cristae can exist transiently as isolated vesicles which are able to fuse and split with other cristae or the IBM (Kondadi et al, 2020a). It was recently shown that OPA1 and YME1L also affect cristae dynamics (Hu et al, 2020). However, it is unclear which metabolic factors are required to ensure cristae membrane dynamics, a process likely to consume considerably amounts of energy, for example, from ATP hydrolysis.

We asked for the bioenergetic parameters which modulate the rates of cristae merging and splitting events: whether ATP levels or the mitochondrial membrane potential ($\Delta\Psi_m$) influence cristae membrane dynamics and to which extent. In this endeavor, we performed advanced live-cell STED SR nanoscopy on mitochondria after inhibition of the electron transport chain (ETC) or the $F_1F_O$ ATP synthase using classical oxidative phosphorylation (OXPHOS) in-hibitors. Furthermore, we used an OXPHOS uncoupler dissipating the $\Delta\Psi_m$. Consistent with earlier studies using EM, we observed ~50% mitochondria to be enlarged, which showed decreased cristae density when compared with mitochondria which were not enlarged. We could further dissect and show that enlarged mito-chondria in particular showed a moderate nonsignificant trend of increased cristae membrane dynamics. We conclude that the rate of cristae membrane dynamics is not negatively affected by inhibiting OXPHOS including dissipation of the $\Delta\Psi_m$, reducing mi-tochondrial ATP levels but is rather moderately enhanced in en-larged mitochondria with reduced cristae density. Thus, cristae dynamic events are ongoing despite reduced cristae density. This would be consistent with the view that cristae dynamics is either limited by structural constraints such as densely packed cristae or that reduction in cristae density is followed by an increased cristae fusion and fission rate as kind of a compensatory mechanism. Furthermore, inhibition of adenine nucleotide translocator (ANT) by applying bongkrekic acid (BKA) to HeLa cells led to aberrant cristae morphology. Contrary to our observations using other OXPHOS inhibitors, we observed a clear reduction in cristae membrane dy-namics in a subset of mitochondria, namely those showing aberrant cristae morphology. Overall, our results indicate that cristae mem-brane dynamics is linked to the bioenergetic state of mitochondria and point to a prominent role of the ADP/ATP metabolite exchange in this process.

# Results

## Cristae membrane dynamics is not impaired in mammalian cells treated with OXPHOS inhibitors

Recent application of novel SR techniques has revealed MICOS-dependent intramitochondrial cristae membrane dynamics in living cells (Kondadi et al, 2020a). However, the bioenergetic requirements which define such highly dynamic membrane remodeling processes are unknown. Cristae membrane remodeling has been used to de-scribe cristae dynamic events (i.e., cristae merging and splitting) and overall changes in cristae morphology within a single mitochondrion in this article. Here, we used live-cell STED SR nanoscopy and de-termined dynamic alterations in cristae structure upon inhibition of ETC complexes, the $F_1F_O$ ATP synthase or the $\Delta\Psi_m$. For this, we treated HeLa cells with the following classical drugs: rotenone, antimycin A, oligomycin A inhibiting Complex I, Complex III, Complex V ($F_1F_O$ ATP synthase), respectively, and CCCP, a protonophore dissipating the $\Delta\Psi_m$. We refer to these drugs collectively as mitochondrial toxins throughout the article. We performed respirometry experiments using HeLa cells to validate the function of various mitochondrial toxins we used while imaging (Fig S1). Using respirometry experiments, the basal, maximal respiratory, and spare respiratory capacities were similar to another set of standard mitochondrial toxins commercially available confirming that the mitochondrial toxins we used are functioning as expected (Fig S1A and B). Previously, we employed ATP5I-SNAP, marking $F_1F_O$ ATP synthase, to visualize cristae using live-cell STED SR nanoscopy in living cells (Kondadi et al, 2020a). Therefore, HeLa cells expressing ATP5I-SNAP were treated with silicon–rhodamine (SiR) dye, which binds covalently to the SNAP-tag, followed by addition of mitochondrial toxins. SiR is suitable for SR imaging owing to its photostability and minimal fluorophore bleaching (Lukinavicius et al, 2013). We decided to follow cristae structure and cristae dynamics at very early time points, within 30 min, after addition of the respective toxins because of the following reasons: (1) to determine the im-mediate effect of acute bioenergetic alterations induced by different mitochondrial toxins on cristae remodeling; (2) to minimize secondary effects occurring later such as mitochondrial frag-mentation which imposes methodological limitations for sub-sequent analyses; and (3) cells do not develop any obvious signs of cell death or undergo apoptosis for at least 1 h after the

addition of these mitochondrial toxins according to earlier reports (Minamikawa et al, 1999; Duvezin-Caubet et al, 2006). The concentrations of mitochondrial toxins used in this article are broadly in the range used for real-time respirometry measurements where oxygen consumption has been shown to either increase or decrease depending on the mode of action of mitochondrial toxins used (Kondadi et al, 2020a; Stephan et al, 2020).

Using live-cell STED SR imaging, we followed cristae dynamics of cells treated with or without various mitochondrial toxins (Fig 1A). We achieved an image acquisition time of 0.94 s/frame which is improved compared with 1.2–2.5 s/frame achieved earlier (Kondadi et al, 2020a) using this technique. In all cases, we observed that cristae showed robust dynamic events at a timescale of seconds independent of the presence of any mitochondrial toxin (Figs 1A and S2 and Video 1, Video 2, Video 3, Video 4, and Video 5). Within the time span of each movie, we observed that cristae dynamics constantly revealed the formation and reshaping of X- and Y-like structures observed in our previous study (Kondadi et al, 2020a). There was no apparent difference in the way cristae appeared or remodeled between the different toxins. Yet to analyze this in more detail and to test whether we missed subtle changes, we performed a blind quantification of cristae merging and splitting events within individual mitochondria. In our previous study, we obtained a spatial resolution of 50–60 nm using live-cell STED nanoscopy (Kondadi et al, 2020a) meaning that cristae with a distance more than 60 nm between them can be distinguished. We found that treatment of cells with mitochondrial toxins did not lead to any significant changes in the frequency of merging or splitting events (simplified depiction shown in Fig 1B) in mitochondria (Fig 1C). We conclude that inhibition of mitochondrial OXPHOS complexes by these toxins does not impair cristae dynamics or cause an imbalance of merging and splitting events.

### Cristae structure is altered in a subset of mammalian cells treated with mitochondrial toxins

To further analyze whether alterations in cristae dynamics is eventually linked only to a subset of mitochondria, we performed a detailed characterization of cristae architecture upon treatment with various toxins. We noted that the treatments of HeLa cells with various mitochondrial toxins led to the formation of mitochondria with increased width in a fraction of cells (Fig 2A, bottom panel), whereas another fraction did not show a change in mitochondrial width (Fig 2A, top panel) and resembled control cells. We categorized the percentage of mitochondria possessing corresponding mitochondrial widths under each condition (Fig 2B–F). Frequency distribution curves revealed that control cells displayed a Gaussian-like distribution of mitochondrial width with the highest percentage of mitochondria present in 400–500 nm range, whereas a maximum mitochondrial width of 600 nm was observed (Fig 2B). On the contrary, treatment of cells with mitochondrial toxins led to substantially increased mitochondrial width (Fig 2A, bottom panel and Fig 2C–F) as shown by a shift towards the right in percentage mitochondria. We found that irrespective of the toxin used, around 50% of the mitochondria (maximum two mitochondria considered per cell) were enlarged (width ≥ 650 nm), and that no mitochondria under control conditions had an average

width larger than or equal to 650 nm (Fig 2B–F). Hence, mitochondrial dysfunction induced by rotenone, antimycin A, oligomycin A, and CCCP uniformly led to enlarged mitochondria within 30 min. Based on these results, we used 650 nm mitochondrial width as the cut-off for defining mitochondria as "enlarged" (from here on) as this excluded all the mitochondria from the control group (referred to as "normal" mitochondria here on). We next quantified cristae structure-related parameters for all mitochondria including distributing them into subsets of normal and enlarged mitochondria. We characterized the cristae number per $\mu m^2$ of mitochondrial area defined as cristae density, average distance between cristae in a mitochondrion defined as intercristae distance and the percentage area occupied by cristae within a mitochondrion. We did not find major differences in different cristae parameters described above when we compared the entire population of mitochondria in cells treated with or without various mitochondrial toxins (Figs 2G and S3A and C). Still, we observed an apparent trend indicating that cristae density is negatively correlated with mitochondrial width when mitochondrial toxins were applied (Fig S3E–I). Control HeLa cells exhibited a median cristae density of around 7 cristae/$\mu m^2$ which was similar to cells treated with mitochondrial toxins (Fig 2G). When we distributed the mitochondria as having normal or enlarged width, we found that mitochondria with normal mitochondrial width showed similar cristae density compared with untreated mitochondria (Fig 2H). Enlarged mitochondria exposed to mitochondrial toxins had a median cristae density of 4 cristae/$\mu m^2$ compared with 7 cristae/$\mu m^2$ in control cells. Only mitochondria showing enlarged width showed a statistically significant decrease of the cristae density for all toxins when compared with control mitochondria (Fig 2H). Thus, these results indicate that reduced cristae density is an effect of mitochondrial enlargement upon application of mitochondrial toxins. These findings are well recapitulated by the observed increased trend in the average intercristae distance which is altered again for enlarged mitochondria (Fig S3A and B). We next checked whether applying mitochondrial toxins led to a change in the percentage cristae area occupied per mitochondrion. We observed that the percentage cristae area per mitochondrion was unchanged upon addition of mitochondrial toxins within the time window of imaging (Fig S3C) which was independent of the mitochondrial width (Fig S3D). Taken together, STED SR nanoscopy revealed that bioenergetically compromised mammalian cells within a short time span result in structural changes where ~50% of mitochondria are characterized by decreased cristae density, increased average distances between adjacent cristae with no gross changes in the relative cristae area occupied by mitochondria. These observations are reflected in the negative correlation of cristae density and mitochondrial width (Fig S3E–I). We next aimed to check ultrastructural changes under these conditions using EM. Consistent with results obtained using STED nanoscopy (Figs 2A–F and S3), electron micrographs revealed enlarged mitochondria and increased distance between the cristae (shown using white arrows) upon treatment of HeLa cells with all mitochondrial toxins (Fig 2I). Increased distances between the cristae contributed to a visible decrease in the cristae density compared with control mitochondria which was in line with previous observations that

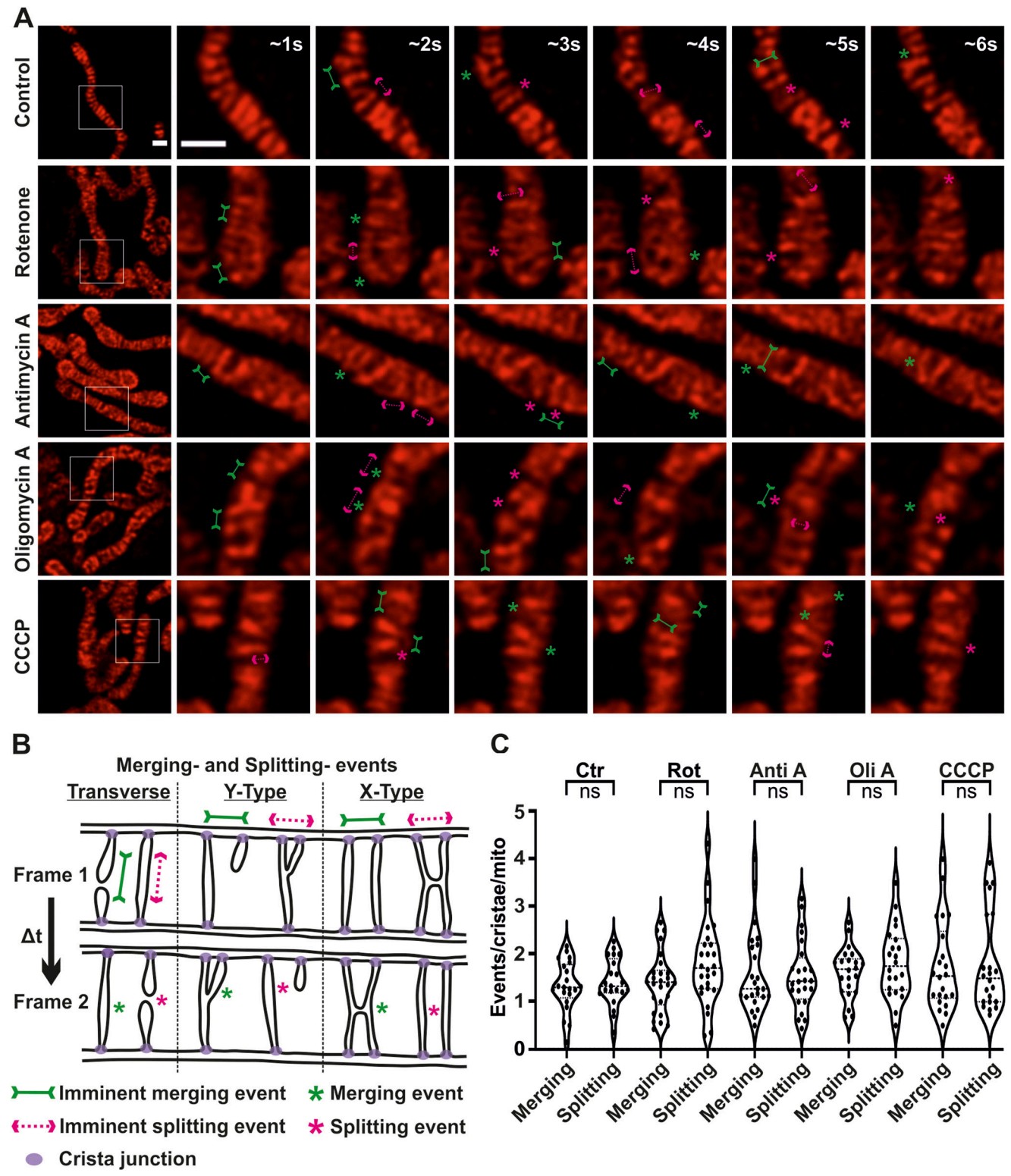

**Figure 1. Crista merging and splitting events occur in a balanced manner upon inhibition of OXPHOS complexes and ΔΨm dissipation.**
**(A)** Representative live-cell STED SR images of HeLa cells, expressing ATP5I-SNAP and stained with silicon–rhodamine, untreated or treated with rotenone, antimycin A, oligomycin A, and CCCP. Images at the extreme left show whole mitochondria along with white inset boxes. Other images on the right-side display time-lapse series (0.94 s/frame) of zoom of mitochondrial portion in the white inset at ~1, 2, 3, 4, 5, and 6 s. Green and magenta asterisks show corresponding merging and splitting events, whereas solid green arrows pointing inward and dotted magenta arrows pointing outward show imminent merging and splitting events, respectively. Scale bar represents 500 nm. **(B)** A scheme illustrating cristae merging and splitting events is shown. **(A, C)** Blind quantification of cristae merging and splitting events per mitochondrion in

cells treated with different mitochondrial toxins resulted in enlarged mitochondria accompanied by decreased cristae density (Gottlieb et al, 2003; Hytti et al, 2019). Overall, using a combination of STED SR nanoscopy and EM, we show that treatment of HeLa cells with various mitochondrial toxins, which disrupt the ETC function and $\Delta\Psi_m$, resulted in enlarged mitochondria accompanied by increased intercristae distance and reduced cristae density.

## Cristae membrane dynamics is unchanged in enlarged mitochondria treated with various mitochondrial toxins

Given the structural alterations in a subset of mitochondria and the finding that cristae dynamics is overall robustly occurring in bioenergetically compromised mitochondria (Fig 1), we wondered whether cristae dynamics is specifically altered in mitochondria that have been structurally altered and the overall effect was masked. Upon treatment with mitochondrial toxins, in enlarged mitochondria, the frequencies of merging and splitting events remained balanced and we observed X- and Y-like structures appearing and disappearing at a timescale of seconds (Figs 3A and S4A–C and Video 6, Video 7, Video 8, and Video 9). Overall, when we revisited normal and enlarged mitochondria separately, we observed an apparent, yet no significant increase in the frequency of both merging and splitting events in enlarged mitochondria after cells were treated with antimycin A and oligomycin A but not in normal mitochondria after the same treatments (Fig 3A–E). However, in enlarged mitochondria, a significant increase of splitting events was observed after rotenone treatment and of merging and splitting events when the $\Delta\Psi_m$ was dissipated by CCCP (Fig 3C and E). Also, we rule out that cristae membrane dynamics is reduced upon inhibition of OXPHOS in the subset of enlarged mitochondria. In contrast, cristae membrane dynamics is moderately increased after dissipating the $\Delta\Psi_m$. Although all the imaging experiments involving mitochondrial toxins were performed within 30 min, we were interested to understand what happens to cristae morphology when they are treated with respective mitochondrial toxins for longer periods of time. We performed STED SR imaging of mitochondria to visualize the cristae around 4 h after treatment with mitochondrial toxins and found that, in general, there was clear mitochondrial fragmentation. Cristae were clearly interconnected in swollen mitochondria. Thus, imaging mitochondria after longer exposure to mitochondrial toxins is not optimal for studying cristae dynamics which was severely stunted (Fig S5).

To check how the mitochondrial ATP levels were influenced by various mitochondrial toxins and whether there was a correlation between cristae membrane dynamics and ATP levels, we checked the mitochondrial ATP levels after the respective treatments within the time span of 30 min similar to STED nanoscopy. For determining mitochondrial ATP levels, we took advantage of the mitGO-Ateam2 probe (Nakano et al, 2011). mitGO-ATeam2 is a ratiometric intramolecular FRET probe which binds ATP to bring the GFP, acting as FRET donor, close to orange fluorescent protein, the FRET acceptor,

leading to an increased acceptor emission. Hence, reduction of ATP levels leads to decrease in the ratio of emission maximum at 580 nm (orange fluorescent protein)/520 (GFP) nm. mitGO-ATeam2 and mitoAT1.03 probes have been used to study spatiotemporal modulations of mitochondrial ATP levels (Imamura et al, 2009; Nakano et al, 2011). Pseudocolor ratiometric rainbow LUT images clearly showed that emission of 580/520 nm significantly decreased in mitochondria of cells treated with rotenone, antimycin A, and oligomycin A when compared with control HeLa cells (Fig 4A, bottommost panel). Accordingly, quantification of the ratio of emission maximum at 580/520 nm showed that all cells treated with rotenone, antimycin A, and oligomycin A displayed a significant reduction of ATP levels (Fig 4B). Surprisingly, treatment of HeLa cells with CCCP did not show any change in the ATP levels within the short time span of 30 min. Furthermore, we checked whether the reduction of ATP levels was affected by mitochondrial width. For this, we compared the ATP levels of cellular population treated with OXPHOS inhibitors. Cells were binned as having either normal or enlarged mitochondria. However, we did not find any differences in ATP levels of cells which predominantly contained either normal or enlarged mitochondria (Fig 4C). This suggests that the observed reduction in mitochondrial ATP levels in cells treated with mitochondrial toxins precedes formation of enlarged mitochondria. Overall, we conclude that unaltered cristae dynamics in enlarged mitochondria is not because of delayed or inefficient action of mitochondrial toxins demonstrating that cristae dynamics is robustly maintained at reduced ATP levels.

Next, we addressed how $\Delta\Psi_m$ is influenced after applying the respective mitochondrial toxins in the time window which was used for STED SR nanoscopy and ratiometric FRET-based ATP level detection. Antimycin A and CCCP strongly decreased $\Delta\Psi_m$ (Fig S6A). Accordingly, detailed quantification revealed a significant decrease of $\Delta\Psi_m$ when cells were treated either with antimycin A or CCCP when compared with untreated mitochondria (Fig S6B). There was a modest but significant decrease of $\Delta\Psi_m$ with rotenone treatment, whereas cells treated with oligomycin showed no change in the $\Delta\Psi_m$. When we put the rate of cristae dynamics (Fig 3) in the context of $\Delta\Psi_m$ measurements (Fig S6), we find that there was no reduction in the rate of cristae merging and splitting events in enlarged mitochondria upon introduction of rotenone, antimycin A, and CCCP compared with control mitochondria despite a significant decrease of $\Delta\Psi_m$. On the contrary, cells treated with $F_1F_O$ ATP synthase inhibitor, oligomycin A, exhibited no significant change in the number of merging and splitting events compared with control mitochondria and did not show a decrease of $\Delta\Psi_m$ despite a significant reduction in ATP levels. Overall, we conclude that the merging and splitting events occur in enlarged mitochondria independent of $\Delta\Psi_m$. Thus, cristae dynamics appears to operate independent of the $\Delta\Psi_m$ and is maintained even at reduced ATP levels. Furthermore, we also checked if there is any correlation between OPA1 forms and cristae dynamics. It is well known that, at steady state, the long

different conditions as described in (A). Pooled data from three individual experiments with 21–26 mitochondria are shown as violin plots with individual data points. Each symbol represents one mitochondrion. (ns = nonsignificant *P*-value > 0.05). To increase the number of cells considered for quantification, a maximum of two mitochondria were randomly considered from a single cell, throughout the article, where STED nanoscopy was performed. One-way ANOVA was used for statistical analysis.

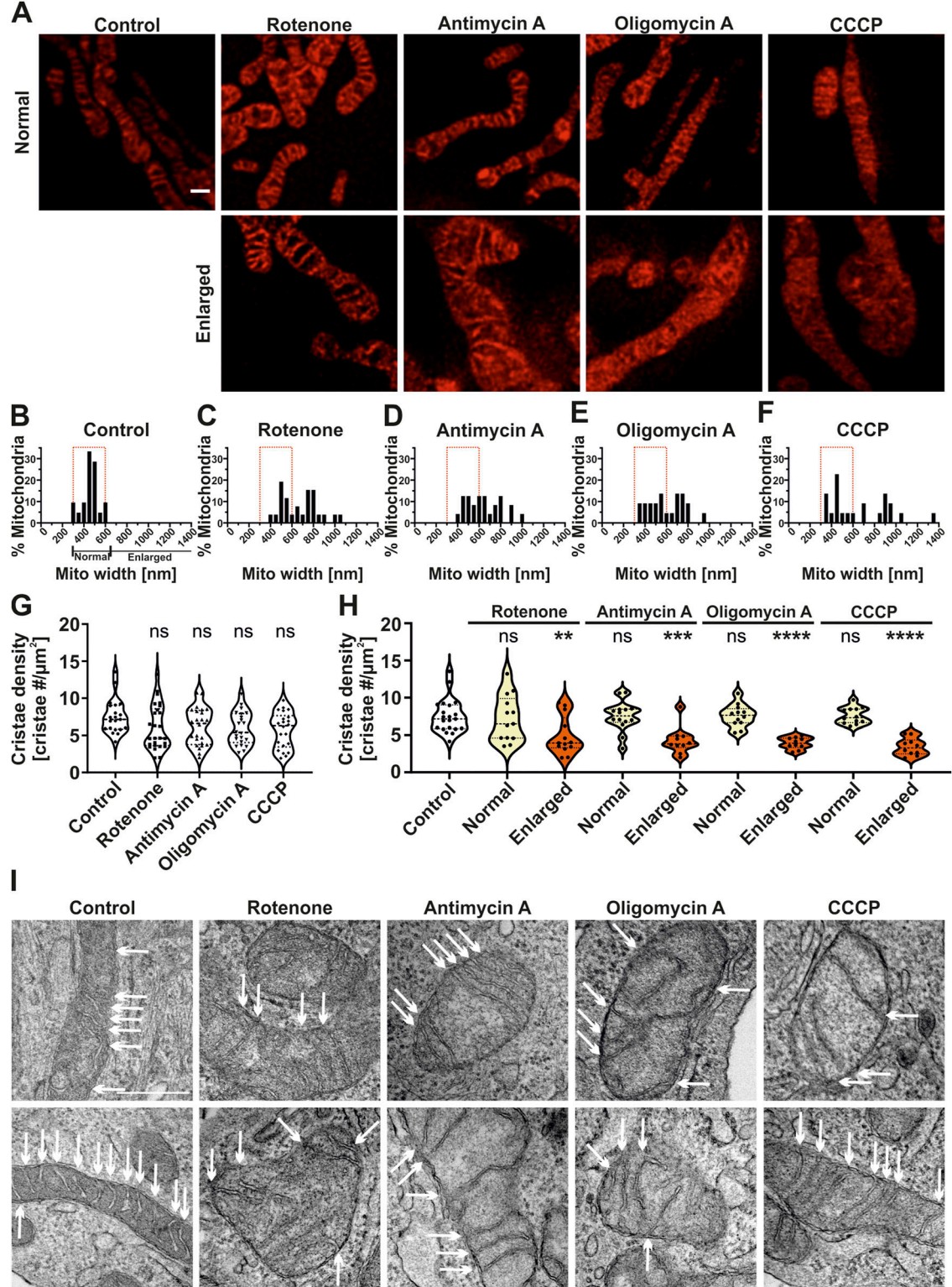

**Figure 2. Mitochondrial toxins alter the morphology of cristae and mitochondria.**
**(A)** Representative STED SR images of HeLa cells expressing ATP5I-SNAP, stained with silicon–rhodamine, displaying normal (<650 nm) or enlarged (≥650 nm) mitochondrial width upon rotenone, antimycin A, oligomycin A, and CCCP treatment. Top and bottom rows show mitochondria with normal and enlarged width, respectively. Scale bar represents 500 nm. **(B, C, D, E, F)** Frequency distribution (50 nm bins) of percentage mitochondria having particular mitochondrial width in control cells (B) and cells treated with rotenone (C), antimycin A (D), oligomycin A (E), and CCCP (F) obtained from three independent experiments (21–26 mitochondria). Red rectangle indicates width distribution of untreated control group which was superimposed in toxin-treated conditions. **(G, H)** Quantification of cristae density (cristae

forms of OPA1 (L-OPA1) are proteolytically cleaved into short forms (S-OPA1) in a balanced manner (Deshwal et al, 2020). Depolarisation of mitochondria leads to conversion of L-OPA1 to S-OPA1 (Duvezin-Caubet et al, 2006; Baker et al, 2014). However, this conversion depends on the concentration of mitochondrial toxins used and treatment time. Thus, we checked if there is any difference in the pattern of L-OPA1 and S-OPA1 in our conditions at 30 min. We found that CCCP treatment leads to enhanced cleavage of L-OPA1 to S-OPA1. There was no difference in patterns of OPA1 forms upon treating with other mitochondrial toxins. Therefore, enhanced merging and splitting events upon CCCP treatment (Fig 3C and E) correlated to accumulation of S-OPA1 (Fig S6C).

### Cristae morphology is perturbed when HeLa cells are treated with an inhibitor of the ANT

We next asked whether ADP/ATP exchange of mitochondria, mediated by the ANT, is regulating the dynamics of cristae membranes. In this context, we used various concentrations of BKA, an ANT inhibitor, at 10, 25, and 50 $\mu$M along with a combination of various mitochondrial toxins, employed while imaging, on HeLa cells (Fig 5A and B). A clear dose-dependent decrease of oxygen consumption was observed with increasing concentration of BKA within the time window (30 min) used for imaging cristae membrane dynamics and ATP levels throughout this article (Fig 5A and B). We used the highest concentration (50 $\mu$M) of BKA for imaging cristae morphology and dynamics as it showed the strongest decrease in mitochondrial oxygen consumption termed BKA response. The maximal respiration was also significantly reduced after addition of different concentrations of BKA when compared with untreated condition. We noted that cristae morphology was clearly aberrant upon BKA treatment compared with untreated controls (Fig 5C and D). STED nanoscopy revealed numerous mitochondria with huge spaces devoid of cristae and highly interconnected cristae upon BKA treatment (Fig 5C and D, top panel, E and Fig S7A and B) and also other abnormal cristae organization where cristae were either clumped or accumulated in the central region of swollen mitochondria (Fig S7C). Overall, the alterations in cristae morphology observed by STED imaging were validated when EM was employed (Fig 5C and D, bottom panel). Therefore, we characterized the percentage of mitochondria having normal and abnormal cristae morphologies. There was a clear increase of mitochondria which had abnormal cristae morphology in BKA-treated condition when compared with untreated cells when the data from all five experiments were pooled as ~33% mitochondria had abnormal cristae morphology when compared with ~13% mitochondria in untreated conditions (Fig S7A and B). There were instances where live-cell STED movies of BKA-treated mitochondria showed highly interconnected cristae where the cristae dynamics was apparently highly reduced or static (Fig 5E and Video 10). We quantified the cristae dynamics in control and BKA-treated mitochondria and found no change in the overall merging and splitting events (Fig S7D). However, BKA-treated mitochondria with abnormal cristae morphology showed significantly reduced cristae dynamics compared with all mitochondria with BKA treatment or untreated mitochondria (Fig 5F and G). The cristae merging and splitting events were still balanced in control cells and BKA-treated cells (Fig S7D). The $\Delta\Psi_m$ was significantly decreased in BKA-treated cells (Fig S7E and F). Thus, despite the overall decrease in $\Delta\Psi_m$, a reduction in cristae dynamics was observed only in those mitochondria where the cristae morphology was aberrant. Overall, we conclude that inhibition of the ANT by BKA results in alteration of cristae morphology and a partial reduction of cristae membrane dynamics suggesting that ADP/ATP exchange across the inner membrane is critical to maintain cristae membrane dynamics independent of the membrane potential.

## Discussion

The development of SR and high-resolution techniques which overcame the diffraction barrier of light, and their recent application to biological structures like mitochondria in fixed and living cells, has opened up exciting prospects to decipher mechanistic insights (Kondadi et al, 2020b; Jakobs et al, 2020). Whereas EM could provide valuable insights into cristae morphology by providing static data at different time-points, one could apply live-cell SR techniques like STED nanoscopy to understand the role of various proteins and metabolic factors regulating mitochondrial cristae dynamics. Here, we asked a basic question, namely whether modulation of OXPHOS, $\Delta\Psi_m$, ATP levels or ADP/ATP exchange in mitochondria determines cristae membrane dynamics, and if so, to which extent. In this study, we used advanced live-cell STED nanoscopy combined with newly developed and optimized quantification methods to study cristae morphology and dynamics when we inhibited the functioning of OXPHOS complexes I, III, V, and dissipated the $\Delta\Psi_m$. Application of a set of well-characterized mitochondrial toxins led to the formation of enlarged mitochondria, yet, contrary to our expectations, none of these toxins blocked cristae membrane dynamics. Before we discuss the details of the latter aspect, it is worth discussing the morphological alterations. Mitochondrial swelling is a phenomenon where there is an increase in the volume of the matrix caused because of osmotic imbalance between the matrix and cytosol (Kaasik et al, 2007). The osmotic balance is regulated by various channels and ion exchangers. Therefore, dysregulation of specific channels and exchangers in mitochondria could result in mitochondrial swelling. In addition, opening of the mitochondrial permeability transition pore causes mitochondrial swelling as the IM becomes permeable to

---

number per mitochondrial area in $\mu$m$^2$) per mitochondria, (G) Pooled data from three individual experiments are shown as violin plots with individual data points (21–26 mitochondria). Each symbol represents one mitochondrion. One-way ANOVA was used for statistical analysis. **(H)** Data were separated into normal and enlarged based on mitochondrial width with each condition having 10–21 mitochondria. Conditions were compared with untreated control group. (ns = nonsignificant $P$-value > 0.05, **$P$-value ≤ 0.01, ***$P$-value ≤ 0.001, **** $P$-value ≤ 0.0001). One-way ANOVA was used for statistical analysis. **(I)** Representative transmission electron micrographs of mitochondria of cells treated without or with rotenone, antimycin A, oligomycin A, and CCCP. Individual cristae within a mitochondrion are marked using white arrows. A higher number of arrows in control mitochondria show increased number of cristae per mitochondrial section when compared with mitochondria where the cells were treated with various mitochondrial toxins. Two mitochondria are shown per condition. Scale bar represents 500 nm.

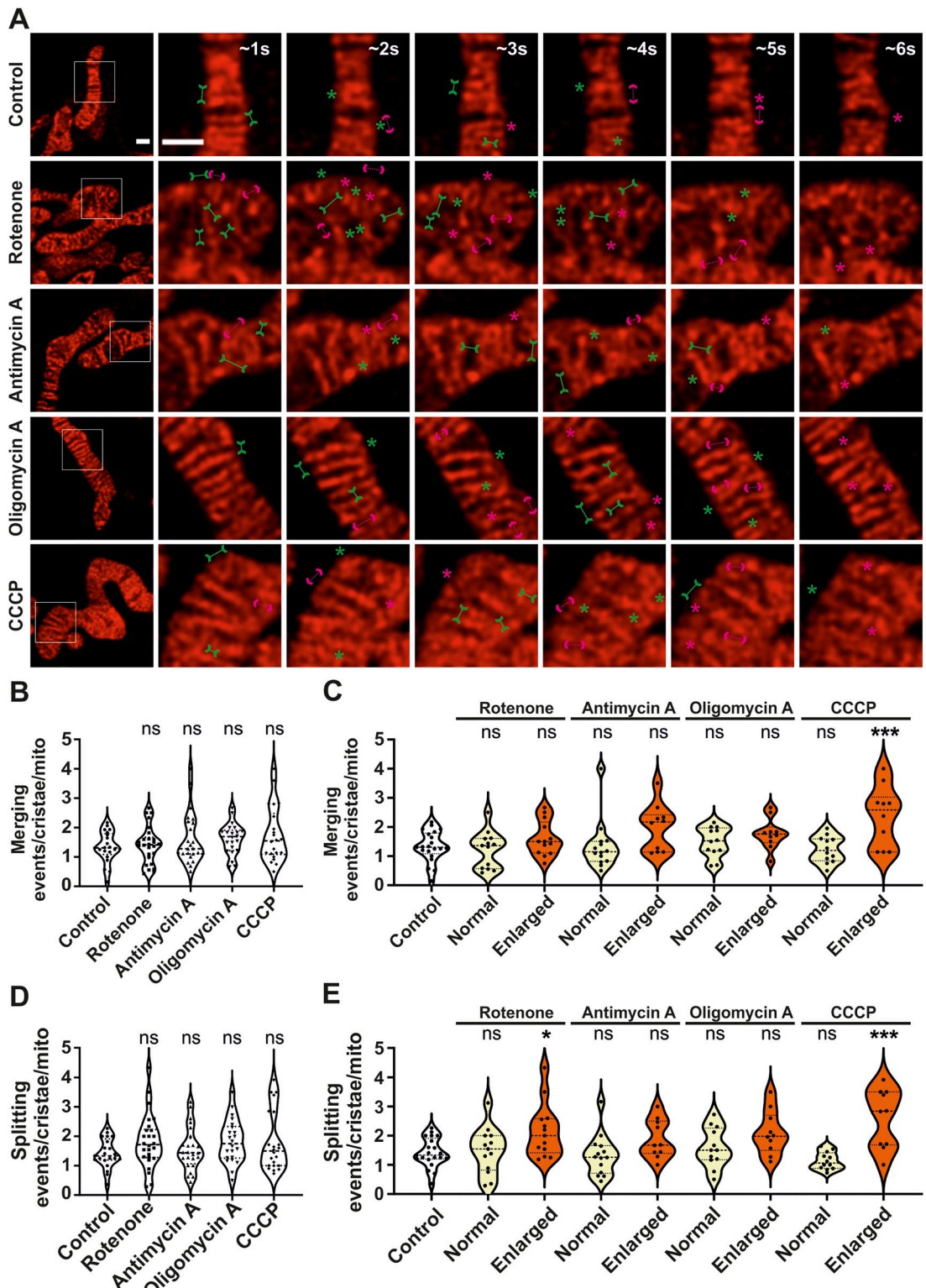

**Figure 3. Crista merging and splitting events are maintained in enlarged mitochondria.**
**(A)** Representative live-cell STED SR images of HeLa cells, expressing ATP5I-SNAP and stained with silicon–rhodamine, showing control and enlarged mitochondria obtained after treatment without or with various mitochondrial toxins respectively. Images at the extreme left show whole mitochondria along with white inset boxes. Other images on the right side display time-lapse series (0.94 s/frame) of zoom of the mitochondrial portion at ~1, 2, 3, 4, 5, and 6 s. Green and magenta asterisks show corresponding merging and splitting events, whereas solid green arrows pointing inward and dotted magenta arrows pointing outward show imminent merging and splitting events, respectively. Scale bar represents 500 nm. **(A, B, C, D, E)** Blind quantification of cristae merging and splitting events per mitochondrion in different

solutes with a molecular weight less than 1.5 kD (Lemasters et al, 2009). Mitochondrial swelling was proposed as mild reversible and excessive irreversible with the former regulating mitochondrial metabolism and the latter leading to mitochondrial dysfunction (Bernardi, 1999; Khmelinskii & Makarov, 2021a, 2021b). The treatment of cells with mitochondrial toxins and imaging within a time window of 30 min using live-cell STED nanoscopy suggests that the mitochondrial enlargement is in reversible mode with no loss of the outer membrane which is consistent with our EM images. EM data from previous studies (Gottlieb et al, 2003; Hytti et al, 2019) are consistent with our live-cell STED nanoscopy and EM observations where the application of the described mitochondrial toxins led to structural alterations in enlarged mitochondria characterized by decreased cristae density. Consistent with our observations, it was shown that dissipation of $\Delta\Psi_m$ by CCCP treatment led to decreased cristae density (Segawa et al, 2020). Concurrent to decreased cristae density, there was a trend of increased intercristae distance which was significantly higher in enlarged mitochondria after treatment with rotenone and CCCP. Overall, the cristae area was not changed when enlarged mitochondria were compared with normal mitochondria treated with mitochondrial toxins or not. Therefore, cristae density was reduced because of an overall increase in the mitochondrial area but not because of changes in the cristae area.

Using live-cell respirometry and consistent with textbooks, it has been shown that mammalian cells instantaneously display decreased oxygen consumption upon inhibition of OXPHOS complexes I, III, and V and increased oxygen consumption upon dissipation of $\Delta\Psi_m$ using CCCP (Kondadi et al, 2020a; Stephan et al, 2020). Thus, addition of various mitochondrial toxins leads to opposing trends of oxygen consumption with CCCP displaying increased mitochondrial consumption as opposed to other three toxins. It is noteworthy to mention that only upon CCCP treatment in enlarged mitochondria, a cleavage of L-OPA1 to S-OPA1 was observed making it tempting to speculate that regulation of cristae merging and splitting events is influenced by accumulation of S-OPA1. It has been shown that a balance of L-OPA1 and S-OPA1 keep CJs tight (Frezza et al, 2006). Furthermore, it was demonstrated that S-Mgm1 (homolog of human OPA1) has the ability to form helical lattice both on the inside and outside of lipid tubes (Faelber et al, 2019). In addition, it could be either a left- or right-handed helix. Both these properties contribute to exert constricting and pulling forces which were proposed to play important roles not only in inner membrane fusion and fission but also in cristae stabilization. At the level of cristae morphology, it is known that depletion of OPA1 leads to reduced number of cristae and CJs (Kushnareva et al, 2013) and disorganized cristae (Olichon et al, 2003). Accordingly, it has been shown that cristae dynamics is reduced in OPA1 KO cells (Hu et al, 2020). Unexpectedly, when the mitochondrial oxygen consumption was reduced after addition of rotenone, antimycin A, and oligomycin A, we did not observe any change in the number of merging and splitting events in enlarged mitochondria when compared with normal mitochondria (depicted in Fig 6). On the contrary, increased oxygen consumption during CCCP exposure is connected to increased number of cristae merging and splitting events. In addition, we demonstrated that the maintenance of the $\Delta\Psi_m$ is not essential for cristae dynamics. Moreover, despite varying differences in cells treated with mitochondrial toxins w.r.t $\Delta\Psi_m$, it can be concluded that largely no changes in the frequency of cristae dynamics were observed when the effects of different toxins are compared (depicted in Fig 6). Our data not only demonstrate that cristae membrane dynamics is not hampered upon loss of the membrane potential, it even shows an increase in merging and splitting events under these conditions. It should be noted that loss of $\Delta\Psi_m$ is not a requirement for mitochondrial enlargement as cells treated with oligomycin A showed enlarged mitochondria but did not lose $\Delta\Psi_m$. Overall, mitochondrial enlargement was necessary but not sufficient to display enhanced cristae membrane dynamics and these data point to the possibility that conditions of high oxygen consumption, which is equivalent to high electron flow from NADH to oxygen in the respiratory chain, may be one criterion to promote cristae merging and splitting events. Another criterion which has already been introduced is that the cristae dynamics might be regulated by OPA1 cleavage which was only observed in CCCP treatment and not in treatments with other mitochondrial toxins within 30 min. Thus, OPA1 cleavage could be a possible mechanism for regulating cristae dynamics.

What is the functional interplay between ATP levels and cristae dynamics? To decipher the ATP levels at the level of mitochondria, we used mitGO-ATeam2 probe which is a genetically encoded sensor based on FRET for detecting differences in ATP levels (Nakano et al, 2011). The ATP levels are based on ratiometric FRET imaging meaning that the expression levels of the construct do not influence the ATP measurements. Furthermore, it was shown that removal of glucose from the culture media results in a decrease of FRET ratio of mitochondrial ATP levels from 1.0 to ~0.7–0.9 in various cell types (Depaoli et al, 2018). Consistent with this previous study, we found a decrease in ATP levels of mitochondria. Whereas we found a consistent decrease of mitochondrial ATP levels in all cells exposed to mitochondrial toxins (except CCCP), cells containing normal mitochondria already showed decreased ATP levels. Interestingly, there was no further decrease in mitochondrial ATP levels in cells containing enlarged mitochondria indicating that the increased cristae dynamics served to maintain the already reduced ATP levels. It is interesting to note that the cristae merging and splitting was increased in enlarged mitochondria which coincided with maintenance of mitochondrial ATP levels upon CCCP treatment. Next, inhibition of the ANT translocator by BKA treatment led to increased percentage of mitochondria with abnormal cristae morphology (depicted in Fig 6). Overall, when analyzing all mitochondria at first,

conditions described in (A). **(B)** Quantification of cristae merging events per mitochondrion from three individual experiments (21–26 mitochondria) is shown as violin plots with individual data points. Each symbol represents one mitochondrion. **(C)** The number of cristae merging events were classified into normal (<650 nm) or enlarged (≥650 nm) mitochondria, with each condition having 10–21 mitochondria. Mitochondrial toxin treatment conditions were compared with untreated control group. **(D)** Quantification of cristae splitting events per mitochondrion from three individual experiments (21–26 mitochondria) is shown as violin plots with individual data points. Each symbol represents one mitochondrion. **(E)** The number of cristae splitting events were classified into normal (<650 nm) or enlarged (≥650 nm) mitochondria, with each condition having 10–21 mitochondria. Different conditions were compared with the untreated control group. (ns = nonsignificant $P$-value > 0.05, *$P$-value ≤ 0.05, ***$P$-value ≤ 0.001). One-way ANOVA was used for statistical analysis.

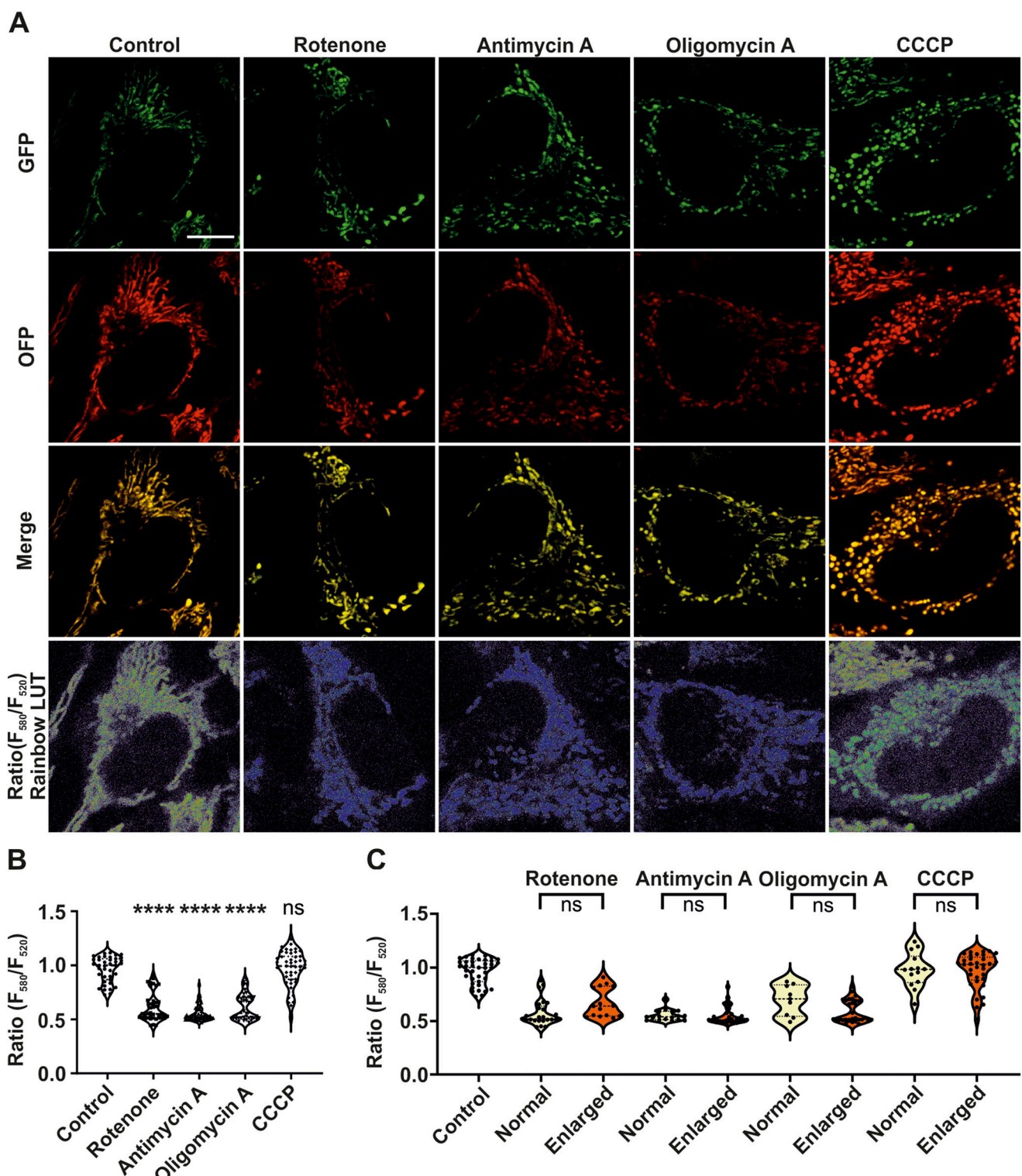

**Figure 4. Mitochondrial ATP levels are significantly reduced upon inhibition of ETC complexes I, III, and V.**
**(A)** Representative images of HeLa cells expressing mitGO-ATeam2, a ratiometric FRET-based genetically-encoded sensor determining the ATP levels, in cells treated without or with rotenone, antimycin A, oligomycin A, and CCCP. The images in first row show the FRET donor emission (GFP), whereas the images in second row display the FRET acceptor emission (OFP). The third row represents a merge of FRET donor and acceptor emission channels. The bottommost row represents ratiometric 32-bit float images, shown using pseudocolour rainbow LUT intensities, used as a basis for quantifying mitochondrial ATP levels. Rainbow LUT intensities reveal low-intensity blue pixels in cells exposed to mitochondrial toxins compared with high-intensity green and red pixels in untreated control cells. Scale bar represents 10 $\mu m$.
**(B, C)** Quantification of cellular mitochondrial ATP levels obtained by dividing the intensities of FRET acceptor emission (580 nm) by FRET donor emission (520 nm) in

we did not observe significant changes in cristae membrane dynamics, yet we detected the subpopulations of mitochondria with apparent different cristae membrane dynamics. When we considered this and divided our population in mitochondria where cristae morphology was abnormal versus normal, the cristae merging and splitting were significantly decreased compared with untreated mitochondria (depicted in Fig 6). This agrees with previous data which showed cristae dynamics was reduced in *MIC13* KO (Kondadi et al, 2020a). Mitochondrial ultrastructure was aberrant in *MIC13* KO because of loss of CJs. Therefore, we propose that cristae morphology and dynamics is interlinked. Given that enlarged mitochondria upon CCCP treatment show enhanced cristae dynamics, we propose that cristae dynamics is possibly determined by structural constraints. In such a scenario, highly, densely packed cristae would impose a constraint limiting cristae dynamics. It may also be that a reduction in cristae density is followed by an increased cristae fusion and fission rate serving as kind of a compensatory mechanism. Another aspect that appears to be important with respect to regulation of cristae membrane dynamics is the possible link to metabolic flux across the inner membrane. As discussed, we observe enhanced cristae membrane dynamics when the $\Delta\Psi_m$ is dissipated resulting in increased oxygen consumption, a condition characterized by high electron and proton flux. We further observe ongoing cristae dynamics even when various OXPHOS inhibitors were applied. It should be emphasized that under these conditions, cellular ATP demand is partially compensated by enhanced glycolysis and that ADP/ATP exchange across the inner membrane is still possible and maintained. This exchange is known to restore $\Delta\Psi_m$ partially by two mechanisms, namely by the electrogenic exchange of cytosolic $ATP^{4-}$ with matrix-located $ADP^{3-}$ and by the reverse $F_1F_O$-ATPase–driven proton-pumping activity requiring the constant import of ATP from the cytosol. Importantly, we observed that under conditions that hamper ADP/ATP exchange cristae membrane dynamics is partially blocked whereas, neither isolated inhibition of OXPHOS complexes I, III, V, nor mild reduction in ATP levels hampered cristae membrane dynamics grossly. We propose that next to structural constraints, in particular, the extent of ADP/ATP flux across the inner membrane is regulating cristae dynamics. Future studies will have to dissect which metabolite fluxes are of particular importance and how they are interconnected. Yet, our study reveals important and partly unexpected insights into the interlink between different modes of OXPHOS modulation and cristae membrane dynamics.

## Materials and Methods

### Cell culture, transfection, and mitochondrial toxin treatment

HeLa cells were maintained in DMEM cell culture media with 1 g/liter glucose (PAN-Biotech), 1 mM sodium pyruvate (Gibco), 2 mM glutaMAX (Gibco), Pen-Strep (PAN-Biotech, penicillin 100 U/ml and streptomycin 100 µg/ml) and 10% FBS (PAN Biotech) at 37°C and 5% $CO_2$. The cells were transfected with 1 µg of ATP5I-SNAP (Kondadi et al, 2020a) or 1 µg of mitGO-ATeam2 plasmid DNA using GeneJuice (Novagen) reagent for 48 h according to the manufacturer's protocol. For live-cell SR imaging, HeLa cells expressing ATP5I-SNAP were stained with 3 µM SNAP-cell 647-SiR (silicon–rhodamine) (NEB) for 30 min, in FluoroBrite DMEM media (Gibco) without phenol red containing 10% FBS (PAN Biotech), 1 mM sodium pyruvate (Gibco), 2 mM glutaMAX (Gibco) and Pen-Strep (penicillin 100 U/ml and streptomycin 100 µg/ml; Sigma-Aldrich). After silicon–rhodamine staining, cells were washed twice with FluoroBrite media. The third wash was done 10 min after the second wash after which mitochondrial toxins were added. Live-cell STED imaging was done in a time window of 10–30 min after addition of toxins at 37°C and 5% $CO_2$. The following concentrations of mitochondrial toxins (Merck) were used: rotenone (5 µM), antimycin A (10 µM), oligomycin A (5 µM), CCCP (10 µM), and BKA (10, 25 or 50 µM). A later time-point involving STED imaging was done in the last 15 min of 4 h.

### Live-cell STED super-resolution nanoscopy and quantification of cristae dynamics

Live-cell STED SR imaging was performed on Leica SP8 laser scanning microscope equipped with a 93X glycerol objective (N.A = 1.3) and a STED module. The samples were excited using a white light laser at 633 nm and the images were collected at emission wavelength from 640 to 730 nm using a hybrid detector (HyD) while using a pulse STED depletion laser beam at a wavelength of 775 nm. To increase the specificity of the signal, gating STED was used from 0.8 ns onwards. An optimised pixel size of 22 nm was used and images were obtained at a rate of 0.94 s/frame. Before every imaging session, the alignment of the excitation and depletion laser beams was optimised using 80-nm colloidal gold particles (BBI Solutions) to ensure the maximum possible resolution. Huygens Deconvolution software (21.10.0p0) was used to process the acquired images. The raw data images are provided. The STED videos were carefully analysed frame-wise and manually quantified in a blind manner to account for the average number of merging and splitting events per cristae within a mitochondrion using ImageJ software (Fiji). The average number of merging and splitting events per mitochondrion was determined and the whole mitochondrial population belonging to a particular condition was represented using violin plots.

### Quantification of various parameters related to cristae morphology

As cellular bioenergetic status influences mitochondrial ultrastructure, the cristae morphology of various mitochondria is fairly

---

HeLa cells treated with or without the mentioned mitochondrial toxins. **(B)** Quantification of mitochondrial ATP levels (ratiometric data) is shown as violin plots from three individual experiments (39–50 cells). Each symbol represents mitochondrial ATP levels of an individual cell. Conditions are compared with untreated control group. **(C)** Ratiometric data were separated into cells with either prevalent normal or enlarged mitochondria (described in methods). Cells with mixed mitochondrial morphology were excluded from this evaluation resulting in 9–39 cells for each group. Statistical analysis was performed between the two classified groups for each treatment condition, with untreated control group as reference. (ns = nonsignificant *P*-value > 0.05, ****P*-value ≤ 0.0001). One-way ANOVA was used for statistical analysis.

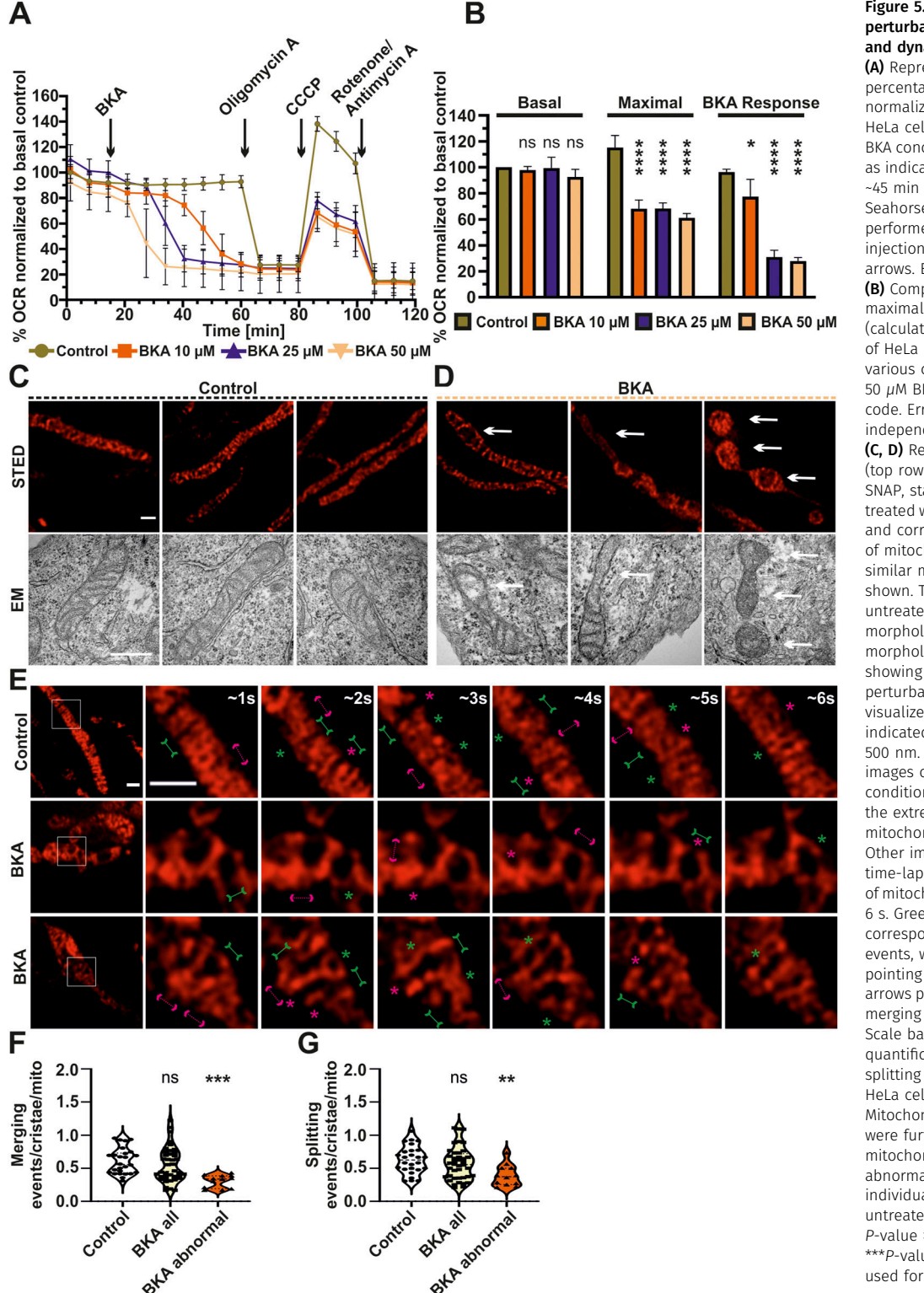

**Figure 5. Inhibition of ANT causes perturbations in cristae morphology and dynamics.**
**(A)** Representative experiment showing percentage oxygen consumption rates, normalized to basal respiration of control HeLa cells, treated without or with various BKA concentrations (10, 25 or 50 μM BKA, as indicated in the color code) are shown. ~45 min after BKA injection, routine Seahorse Mito Stress Test was performed. Respective compound injection time-points are indicated by black arrows. Error bars represent SD. **(B)** Comparison of basal respiration, maximal respiration, and BKA response (calculated ~32 min after BKA injection) of HeLa WT cells treated without or with various concentrations of BKA (10, 25 or 50 μM BKA) as indicated using a color code. Error bars represent SD from three independent biological replicates. **(C, D)** Representative STED SR images (top row) of HeLa cells expressing ATP5I-SNAP, stained with silicon–rhodamine treated without (C) or with (D) 50 μM BKA and corresponding electron micrographs of mitochondria (bottom row) displaying a similar mitochondrial ultrastructure are shown. Three columns (C) display untreated mitochondria with normal morphology (D) Abnormal cristae morphology of BKA-treated mitochondria showing regions of sparse cristae. Similar perturbations in cristae morphology visualized by STED and EM images are indicated by arrows. Scale bars represent 500 nm. **(E)** Additional live-cell STED SR images of HeLa cells, from same conditions as (C, D) are shown. Images at the extreme left show whole mitochondria along with white inset boxes. Other images on the right-side display time-lapse series (0.94 s/frame) of zoom of mitochondrial portion at ~1, 2, 3, 4, 5, and 6 s. Green and magenta asterisks show corresponding merging and splitting events, whereas solid green arrows pointing inward and dotted magenta arrows pointing outward show imminent merging and splitting events, respectively. Scale bar represents 500 nm. **(F, G)** Blind quantification of cristae merging (F) and splitting (G) events per mitochondrion in HeLa cells treated without or with BKA. Mitochondria from BKA-treated cells were further separated into all mitochondria or those with exclusively abnormal cristae morphology and the individual groups compared with the untreated control (ns = nonsignificant $P$-value > 0.05, **$P$-value ≤ 0.01, ***$P$-value ≤ 0.001). One-way ANOVA was used for statistical analysis.

uniform in individual cells. Thus, we maximised the number of cells used for STED nanoscopy by considering only a maximum of two mitochondria from each cell when they were treated with or without toxins, where we quantified various cristae parameters including merging and splitting events. We determined mitochondrial width, cristae density, average intercristae distance, and

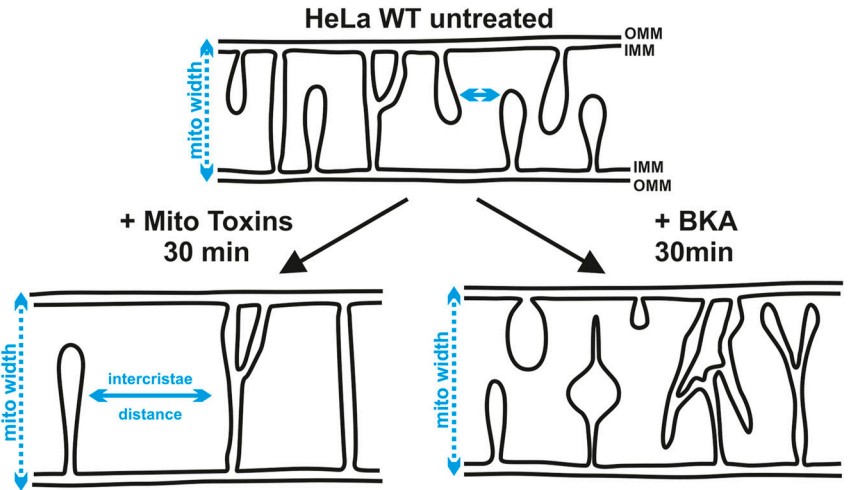

**HeLa WT untreated**

**+ Mito Toxins**
**30 min**

- OXPHOS inhibited
- Cristae morphology <u>lesser</u> effected
- Cristae dynamics normal or partially <u>enhanced</u>

**+ BKA**
**30min**

- ADP/ATP exchange inhibited
- Cristae morphology <u>drastically</u> effected
- Cristae Dynamics <u>significantly</u> <u>reduced</u>

**Figure 6. A model showing the influence of mitochondrial toxins on cristae morphology and dynamics.**
On one hand, treatment of HeLa cells with various mitochondrial toxins (rotenone, antimycin A, oligomycin A, and CCCP) leads to inhibition of ETC or the $F_1F_O$ ATP synthase along with enlargement of mitochondria. The distance between the cristae (intercristae distance) increases because of mitochondrial enlargement, whereas the cristae dynamics is either unchanged or increased (CCCP treatment). On the other hand, BKA treatment, inhibiting ATP/ADP exchange, also leads to mitochondrial enlargement. However, cristae morphology and the number of merging and splitting events are severely reduced in a subset of mitochondria.

percentage cristae area occupied by mitochondria using custom-made macros in Fiji. To determine the width of an individual mitochondrion in images obtained using STED nanoscopy, we used the average of three separate line scans covering the maximum diameter at both ends and the centre of mitochondria which were roughly drawn equidistant from each other. Furthermore, to determine the cristae density, a segmented line was manually drawn across the length of each mitochondrion and an intensity profile of the pixels across the length of the line was created. The algorithm detected the number of cristae by using the number of maximum intensity points of the graph along the length of mitochondrial plot profiles. The obtained cristae number was divided by the previously determined area of the respective mitochondrion to calculate the cristae number per $\mu m^2$ which we termed cristae density.

To measure the average distance between cristae defined as intercristae distance (in nm), we used the previously acquired intensity profiles of mitochondria to determine the exact coordinates of each crista in the image. Euclidean distances between cristae were calculated using a custom-made macro. Because of the drastic variations in mitochondrial and cristae morphology upon BKA treatment, the above-described macro for determination of cristae number was not used. For these datasets, the number of cristae per mitochondrion was counted manually and used for normalization of merging and splitting events. Next, for calculation of the percentage cristae area occupied by mitochondria, we used a semiautomated batch-processing custom-made macro. Cristae structures were manually selected by applying appropriate threshold on the images and the total mitochondrial area was selected by drawing the outline of the whole mitochondrion. The "Analyze Particles" function of Fiji was used to calculate the cristae area where structures less than five pixels were excluded. The macro divided the sum of all cristae area by the whole mitochondrial area and multiplied the result by 100 to acquire the percentage cristae area occupied by that particular mitochondrion.

## Electron microscopy

HeLa cells were grown in 15-cm petri dishes at 37°C with 5% $CO_2$ and treated with respective mitochondrial toxins for 30 min which were then fixed with 3% glutaraldehyde, 0.1 M sodium cacodylate buffer at pH 7.2. Cell pellets were washed in fresh 0.1 M sodium cacodylate buffer at pH 7.2, before embedding in 3% low melting agarose. They were stained by incubating in 1% osmium tetroxide for 50 min followed by two washes for 10 min with 0.1 M sodium cacodylate buffer and one wash with 70% ethanol for 10 min. Samples are stained using 1% uranyl acetate/1% phosphotungstic acid mixture in 70% ethanol for 60 min. Graded ethanol series was used to dehydrate the specimen. The samples were embedded in spur epoxy resin for polymerization at 70°C for 24 h. Ultrathin sections obtained using a microtome were imaged with a transmission electron microscope (H600; Hitachi) at 75 V which had a Bioscan 792 camera (Gatan).

## FRET-based microscopy to measure ATP levels

Cells expressing the genetically-encoded mitGO-ATeam2 were used to determine the ATP levels, kindly provided by Hiromi Imamura, Kyoto, Japan (Nakano et al, 2011). Single optical sections were obtained with a 93X glycerol objective (N.A = 1.3) using Leica SP8 confocal microscope maintained at 37°C and 5% $CO_2$. The samples were excited at 471 nm and the green and orange emission channels were simultaneously obtained from 502 to 538 nm (termed 520 nm) and 568–592 nm (termed 580 nm), respectively, as described (Nakano et al, 2011) in the photon counting mode. To quantify the ratiometric images obtained, a semiautomated custom-made macro was designed using Fiji software to analyse the acquired images in a batch processing mode. The cells of interest were manually selected by drawing a region of interest. The obtained orange emission channel images

(580 nm) were divided by respective green emission channel images (520 nm) by using the "Image Calculator" function of Fiji. A threshold was manually applied on the resulting ratiometric 32-bit float image to exclude background pixels using the "Clear Outside" command. To categorise cell population as containing either swollen or normal mitochondria, the cut off for swollen mitochondria was set to 650 nm in congruence with STED SR nanoscopy. If 85% of the mitochondrial population featured enlarged mitochondria, the cells were designated as swollen. Similarly, if 85% of the mitochondrial population featured mitochondria whose width was less than 650 nm, the cell was considered as having normal mitochondria. We measured the diameter of whole mitochondrial population in the respective cells using Leica Application Suite X software (version 3.7.1.21655).

### Determination of mitochondrial membrane potential ($\Delta\Psi_m$)

HeLa cells were incubated with 20 nM TMRM (Invitrogen) and 50 nM MitoTracker Green (Invitrogen) in DMEM cell culture media along with other supplements (mentioned above) for 30 min at 37°C followed by three washes. 10 min after the addition of respective toxins, cells were imaged for 20 min in DMEM media containing 10 mM HEPES buffer (Gibco) and other supplements. Mitochondrial toxins were present in the media during imaging sessions. Imaging was done on spinning disc confocal microscope (PerkinElmer) using a 60x oil-immersion objective (N.A = 1.49). Single optical sections were obtained using excitation wavelengths of 488 nm (MitoTracker Green) and 561 nm (TMRM). The microscope was equipped with a Hamamatsu C9100 camera. Image analysis including background subtraction and measurement of mean fluorescence intensity were performed using Fiji software after drawing a region of interest around individual cells.

### Live-cell respirometry

All the respiration measurements were performed using Seahorse XFe96 Analyzer (Agilent). HeLa cells were seeded in Seahorse XF96 cell culture plate (Agilent) at a density of $3.5 \times 10^4$ cells per well overnight. Next day, cells were washed and incubated in basic DMEM media (103575-100; Agilent) supplemented with 10 mM glucose (Sigma-Aldrich), 2 mM glutamine (Thermo Fisher Scientific), and 1 mM pyruvate (Gibco) at 37°C, with no $CO_2$ incubation 1 h before the assay. For testing the functionality of mitochondrial toxins we used, they were compared with commercially available Seahorse compounds. Thus, mitochondrial respiration was measured using Seahorse XF Cell Mito Stress Test kit (Agilent) according to the manufacturer's instructions by using rotenone (0.5 $\mu$M), antimycin A (0.5 $\mu$M), oligomycin A (1 $\mu$M), FCCP (0.5 $\mu$M) or using corresponding concentration of mitochondrial toxins used for microscopy experiments described in the mitochondrial toxin treatment methods section.

For the BKA experiment, three concentrations of BKA (Sigma-Aldrich) were tested at 10, 25, and 50 $\mu$M. The dilutions of BKA and all corresponding mitochondrial toxins used throughout the article were prepared in Seahorse medium. The duration between any two measurements is ~6 min. BKA response was calculated ~32 min after BKA injection (fifth measurement after BKA addition).

The measurements after BKA injection were followed by subsequent injections of oligomycin A, CCCP, and a mixture of rotenone and antimycin A as routinely performed to assess mitochondrial oxygen consumption in Seahorse live-cell respirometry experiments. Cell number was normalized after the run using Hoechst staining. Data were analysed using wave software (Agilent). Further calculations were done in Microsoft Excel and figure preparation in GraphPad Prism.

### SDS gel electrophoresis and Western blotting

Cells were treated for 30 min with the respective toxins using same concentrations as in the imaging experiments. Cells were washed thrice with cold DPBS (PAN-biotech) and harvested by scrapping and pelleting at 1,000$g$, 4°C for 10 min. Cell pellets were resuspended in an appropriate volume of RIPA buffer 150 mM NaCl, 0.1% SDS, 0.05% sodium deoxycholate, 1% Triton-X-100, 1 mM EDTA, 1 mM Tris, pH 7.4, 1x protease inhibitor (Sigma-Aldrich). Protein concentration was determined using Lowry assay with the DCTM protein assay kit (BIO-RAD). SDS samples were prepared using Laemmli buffer and subsequent heating at 95°C for 5 min 20 $\mu$g protein were loaded on 10% SDS–PAGE gels. After SDS–PAGE, the proteins were transferred onto a nitrocellulose membrane. To assess loading and transfer quality, the membrane was stained using Ponceau S (Sigma-Aldrich) after the transfer. After 1 h of blocking the membrane with 5% milk in TBS-T at room temperature, it was decorated against OPA1 (Pineda, custom-made) over night at 4°C. Goat IgG anti-rabbit IgG (Dianova) HRP-conjugate was used for detection. The chemiluminescent signals were obtained using Signal Fire ECL reagent (Cell Signaling Technology) and VILBER LOURMAT Fusion SL equipment (Peqlab).

### Statistics and data representation

Statistical significance was tested by one-way ANOVA followed by Dunnett´s test for multiple comparisons against single control group or Šídák's test for multiple comparisons of selected pairs with ns = nonsignificant $P$-value > 0.05, *$P$-value ≤ 0.05, **$P$-value ≤ 0.01, ***$P$-value ≤ 0.001, ****$P$-value ≤ 0.0001. For statistical analysis and data representation, GraphPad Prism (version 9.5.1) was used.

# Supplementary Information

# Acknowledgements

We thank Andrea Borchardt and Tanja Portugall for excellent technical assistance in performing electron microscopy and molecular biological experiments. The STED SR and FRET-based ATP experiments were performed at the Centre for Advanced Imaging (CAi), HHU, Düsseldorf. We are grateful to Hiromi Imamura from Kyoto University, Japan, for providing us with the mitGO-ATeam2 plasmid used for detecting ATP levels. Funding: this work was funded by the Deutsche Forschungsgemeinschaft (DFG) SFB 1208, project B12

(ID 267205415), to AS Reichert and by the DFG grant KO 6519/1-1 to AK Kondadi.

## Author Contributions

M Golombek: data curation, software, formal analysis, validation, investigation, visualization, methodology, and writing—review and editing.
T Tsigaras: software, formal analysis, investigation, and visualization.
Y Schaumkessel: formal analysis and investigation.
S Hänsch: resources, software, validation, and methodology.
S Weidtkamp-Peters: resources and methodology.
R Anand: supervision, validation, investigation, visualization, and writing—review and editing.
AS Reichert: conceptualization, supervision, funding acquisition, project administration, and writing—review and editing.
AK Kondadi: conceptualization, supervision, funding acquisition, investigation, project administration, and writing—original draft, review, and editing.

## Conflict of Interest Statement

The authors declare that they have no conflict of interest.

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
