## [Reviewer comments · Life Science Alliance]

Life Science Alliance

Cristae dynamics are modulated in bioenergetically compromised mitochondria

Mathias Golombek, Thanos Tsigaras, Yulia Schaumkessel, Sebastian Haensch, Stefanie Weidtkamp-Peters, Ruchika Anand, Andreas Reichert, and Arun Kumar Kondadi

DOI: <https://doi.org/10.26508/lsa.202302386>

Corresponding author(s): Arun Kumar Kondadi, Heinrich Heine University Düsseldorf and Andreas Reichert, Heinrich Heine University Düsseldorf

Review Timeline:

Submission Date:	2023-09-20
Editorial Decision:	2023-10-11
Revision Received:	2023-10-27
Editorial Decision:	2023-11-02
Revision Received:	2023-11-03
Accepted:	2023-11-06

Transaction Report:

Please note that the manuscript was reviewed at Review Commons and these reports were taken into account in the decision-making process at *Life Science Alliance*.

Review
COMMONS

Reviews

Review #1

In their manuscript „Live-cell super-resolution nanoscopy reveals modulation of cristae dynamics in bioenergetically compromised mitochondria", Golombek et al. tested the effects of different mitochondrial toxins on cristae dynamics. The main focus of their work lies on live STED imaging, which they use to visualize cristae merging and splitting. They found swelling of mitochondria and reduced cristae density in response to most toxins, but cristae dynamics remained largely unaffected. Depletion of the membrane potential by administration of CCCP increased cristae dynamics, while inhibition of ANT had a negative effect on cristae dynamics at least in a subset of mitochondria.

Major comments

- The authors state that the used concentrations of mitochondrial toxins commonly result in a change in oxygen consumption. While this is believable, it is not guaranteed that the specific chemicals used for the experiments were working properly (freeze/thawing or simply incorrect storage or aliquotation may have an effect on the compounds). This is even more important in the case of results where no significant change after the administration of the toxins is seen. In Figure 5, the authors report no change in membrane potential after oligomycin administration, this is unexpected. I therefore suggest to include a supplementary figure, in which the functionality of the compounds is verified. This could be done by respiratory measurements (e.g. Seahorse). A Mito Stress Test was performed for Figure 6, but this was done using the Seahorse kit chemicals, which were probably different from the chemicals used in the microscopy experiments.
- Figure 1 would benefit from a more detailed description of merging/splitting events. Maybe a cartoon plus a zoomed in image of an exemplary event?
- Could the reduced cristae density be an effect of mitochondrial swelling? It is curious that all toxins appear to have the same effect on mitochondrial architecture. What is the fate of an enlarged mitochondrion over time? Mitophagy? And does the percentage of enlarged mitochondria change with increasing treatment time?
- Figure 4C: How was the mitochondrial width determined in the LSM images? Especially in the perinuclear area it will be difficult to determine this parameter without the super-resolution provided by STED. Was this parameter determined manually for selected mitochondria? In the methods part it says that only a maximum of two mitochondria per cell were analyzed. How were these chosen? Was the process blinded?
- What is the average size of all mitochondria per cell? Is this addressed in Figure 2B or are only analyzed mitochondria included? Please clarify. Were the mitochondria chosen for analysis representative for the respective cell?

Minor comments

1. explain the mt-Go-AT team2, what is GFP (green fluorescent protein) and OTP (?)
2. the graphs show in principle, e.g. Fig.1B, 3B-E show events/mitochondrion as far as I understand, not per cristae.
3. I would recommend changing the legend of the x-axis of Fig.2B-F to mito-width (y-axis could be probability density function, PDF).

Referees cross-commenting

both expert opinions address similar concerns and therefore a revision should be requested
The study is thorough and the experiments and results are well described. Overall, however, it remains a descriptive study and does not provide mechanisms. There is also no discussion of how MMP-dependent proteins, such as Opa1, which was previously studied by the Reichert group, might be affected. For swelling mechanisms, the opening of the mitochondrial permeability transition pore was discussed. This could be tested using inhibitors, but perhaps not within the scope of this publication. Nevertheless, the information provided by the study is of interest to the bioenergetics community and should be made available.

Review #2

Summary:

The authors investigated cristae merging and splitting events using ultra-resolution STED. The goal was to test if

cristae membrane remodeling is dependent on OXPHOS complexes, mitochondrial membrane potential ($\Delta\Psi_m$), and the ADP/ATP nucleotide translocator. To do this the authors utilized several mitochondrial toxins with known mechanisms of action. Interestingly, many changed overall cristae density but did not change the cristae remodeling events. Inhibition of ANT did change cristae morphology and cristae dynamics.

****Major Concerns****

1. Many conclusions and concepts need more clarification. For example, a major take home from the abstract is that various ETC inhibitors and protonophores reduce cristae density but did not change cristae remodeling events. If cristae density is reduced, how can this occur without cristae remodeling events? Remodeling events need to be clearly defined in the introduction and abstract.
2. Other interpretations are also unclear such as how ETC inhibitors which reduce ATP levels did not impact cristate remodeling events, yet inhibiting ATP/ADP exchange did greatly impact this phenomenon. It seems likely that the inhibition of ANT has nothing to do with ATP/ADP exchange since most of the ETC inhibitors no doubt greatly impact overall ATP/ADP exchange. This interpretation needs clarification.
3. Why did the authors wait 30 min to image after the addition of mitochondrial toxins? I would have guessed there is a more rapid change in response to these inhibitors. Is there is a chance he authors missed the most dramatic events?
4. How do these mitochondrial toxins that are known to cause mitochondrial swelling not induce changes in cristate density?
5. It's interesting that inhibition of the ANT translocator by BKA treatment led to increased percentage of mitochondria with abnormal cristae morphology. It's accepted that inhibition of ANT profoundly reduces mitochondrial swelling. Do the authors have any data suggesting that abnormal cristae morphology actually is a mechanism for reducing cell death events such as permeability transition? Did the authors utilize cyclosporin A concomitantly with any of the mitochondrial toxins?
6. Are the authors confident in the data given many of the experiments utilized quantification of 10-20 mitochondria? How are you sure this sampling is sufficient for phenomenon being studied?
7. Figure 4 and 5 merely confirm current dogma and don't really contribute to the overall conclusions and can be moved to supplemental data.
8. It's interesting that BKA dose dependently decreased ATP-linked respiration and all doses limited maximal respiratory capacity. It would be interesting to know if the BKA normal vs. abnormal mitochondria have differential membrane potential?
9. Overall, this is an interesting study and seems appropriately performed but the results and conclusions are unclear. More discussion should include physiological relevance and impact and how this data influences previous work. Some physiological perturbations beyond the mitochondrial toxins and or utilization of genetic models would strengthen the interpretation and overall impact.

****Referees cross-commenting****

Yes, I conclude that given the significant overlap in reviewer comments and general need for clarification of concepts and data that a revision is in order.

Overall, a highly specialized study with audience limited to mitochondriacs. Although, I'll note tis is a hot area of study and there is high interest in the field. Some of the data interpretation is difficult to understand and overall more context is needed to explain the results, impact and relevance. Defining exactly what a cristae remodeling event is and how this differs from cristae density and how the two aren't directly connected is unclear.

1. General Statements [optional]

This section is optional. Insert here any general statements you wish to make about the goal of the study or about the reviews.

This section is mandatory. Please insert a point-by-point reply describing the revisions that were already carried out and included in the transferred manuscript.

Response to Reviewers

Reviewer #1 (Evidence, reproducibility and clarity (Required)):

In their manuscript „Live-cell super-resolution nanoscopy reveals modulation of cristae dynamics in bioenergetically compromised mitochondria“, Golombek et al. tested the effects of different mitochondrial toxins on cristae dynamics. The main focus of their work lies on live STED imaging, which they use to visualize cristae merging and splitting. They found swelling of mitochondria and reduced cristae density in response to most toxins, but cristae dynamics remained largely unaffected. Depletion of the membrane potential by administration of CCCP increased cristae dynamics, while inhibition of ANT had a negative effect on cristae dynamics at least in a subset of mitochondria.

<major comments>

1. The authors state that the used concentrations of mitochondrial toxins commonly result in a change in oxygen consumption. While this is believable, it is not guaranteed that the specific chemicals used for the experiments were working properly (freeze/thawing or simply incorrect storage or aliquotation may have an effect on the compounds). This is even more important in the case of results where no significant change after the administration of the toxins is seen. In Figure 5, the authors report no change in membrane potential after oligomycin administration,

this is unexpected. I therefore suggest to include a supplementary figure, in which the functionality of the compounds is verified. This could be done by respiratory measurements (e.g. Seahorse). A Mito Stress Test was performed for Figure 6, but this was done using the Seahorse kit chemicals, which were probably different from the chemicals used in the microscopy experiments.

Response: We appreciate the valid concerns of the reviewer in this point.

A) In order to show the functionality of compounds which were used for performing our experiments including STED imaging, we now performed respiratory measurements employing the concentrations of mitochondrial toxins (Oligomycin A, CCCP, rotenone/antimycin A) which were used during imaging conditions as well as commercially available mitochondrial toxins (Oligomycin A, FCCCP, rotenone/antimycin A) with respective concentrations used as a standard for the Mito stress Kit. The new figures are included in **Fig S1A & B**. HeLa cells treated with seahorse compounds or those used during imaging conditions showed similar results including basal, maximal and spare respiratory capacity. Further, in order to overcome the inefficiency of mitochondrial toxins employed, due to freeze/thaw cycles, we used fresh aliquots (stored at -20°C) as a general strategy. This is clearly observed by a drastic reduction of $\Delta\Psi_m$ upon treating HeLa cells with CCCP, antimycin A as well as rotenone (**Fig S6A & B**). A reduction of mitochondrial ATP levels was also observed upon employing rotenone, antimycin A and oligomycin A confirming that active mitochondrial toxins were used. These experiments demonstrate that the mitochondrial toxins employed throughout our manuscript are functional as expected.

New Figure S1A & B

B) The **Fig 6** (now **Fig 5 due to Reviewer # 2, Point 7**) respirometry experiments which initially employed seahorse compounds and BKA has now been replaced with new experiments where we used mitochondrial toxins similar to STED imaging. Needless, to say, the results are similar

to what were observed with seahorse compounds. The new figures are replaced in **Fig 5A & 5B**.

New Figure 5A & B

C) Oligomycin A inhibits ATP synthase which results in decreased ATP synthesis as observed (**Fig 4A & B**). Further, oligomycin A is expected to hyperpolarise mitochondria (2). In **Fig S6**, despite some cells having more $\Delta\Psi_m$, there was no overall significant change when compared to untreated cells. Previous publications also show that there is no significant difference in $\Delta\Psi_m$ upon treatment with oligomycin (1) demonstrating that the $\Delta\Psi_m$ depends on the concentration of oligomycin, treatment time and cell type.

2. Figure 1 would benefit from a more detailed description of merging/splitting events. Maybe a cartoon plus a zoomed in image of an exemplary event?

Response: Thank you for the suggestion. In order to clearly explain/simplify the understanding of cristae merging and splitting events, we added a cartoon in **Fig 1B**. The green and magenta arrows show sites of imminent merging and splitting with the green and magenta asterisks representing them respectively in the subsequent frames. The zoomed in images in Fig1A (leftmost panel) are shown to the right as time-lapse images.

New Figure 1B

3. Could the reduced cristae density be an effect of mitochondrial swelling? It is curious that all toxins appear to have the same effect on mitochondrial architecture. What is the fate of an enlarged mitochondrion over time? Mitophagy? And does the percentage of enlarged mitochondria change with increasing treatment time?

Response: Thank you for the comment.

A) We agree that the reduced cristae density is due to mitochondrial swelling. We added the relevant text in the results section 'Cristae structure is altered in a subset of mammalian cells treated with mitochondrial toxins'. Treatment of HeLa cells, with all the mitochondrial toxins mentioned, uniformly result around 50 % of mitochondria undergoing enlargement (**Fig 2B**). In enlarged mitochondria where the mitochondrial width is ≥ 650 nm, there is no change in cristae area occupied per mitochondria (**Fig S3C & D**) and as a result reduced cristae density (**Fig 2H**). Therefore, it indicates that reduced cristae density occurs due to mitochondrial enlargement.

Figure 2B-F

Figure S3C and D

B) In order to address the fate of mitochondria with increasing time upon treatment with various mitochondrial toxins, we treated the HeLa cells for 4 hrs with mitochondrial toxins. Untreated cells maintained normal mitochondrial morphology while cells treated with various mitochondrial toxins displayed fragmented and swollen mitochondrial morphology. The new **Fig S5** is included in the supplementary. Cristae morphology was abnormal displaying interconnected cristae in swollen mitochondria. Since mitochondrial fragmentation is already observed at 4 hours and accompanied by interconnected cristae, the number of cristae merging and splitting were severely reduced.

Our imaging performed within 30 mins of addition of respective toxins overcomes the additional aberrancy of mitochondrial fragmentation which would not allow a reliable analysis of cristae dynamics as too few cristae would be visible within one mitochondrion.

New Figure S5

4. Figure 4C: How was the mitochondrial width determined in the LSM images? Especially in the perinuclear area it will be difficult to determine this parameter without the super-resolution provided by STED. Was this parameter determined manually for selected mitochondria? In the methods part it says that only a maximum of two mitochondria per cell were analyzed. How were these chosen? Was the process blinded?

Response: Thank you for the comment. We could imagine the reason for the ambiguity in understanding.

A) For LSM confocal images involving FRET-based microscopy to determine the ATP levels, we calculated the cell population as belonging to either normal or enlarged category. The confocal images of HeLa cells displayed clear separation of mitochondria even in the perinuclear area (representative images are shown in **Fig 4A**) and thus it was possible to measure the width of individual mitochondria. The methods section '**FRET-based microscopy to measure ATP levels**' describes that 'the cut off for swollen mitochondria was set to 650 nm in congruence with STED SR nanoscopy. If 85% of the mitochondrial population featured enlarged mitochondria, the cells were designated as swollen. Similarly, if 85% of the mitochondrial population featured mitochondria whose width was less than 650 nm, the cell was considered as having normal mitochondria'.

Figure 4A

B) The cristae morphology of various mitochondria is fairly uniform in individual cells. Thus, the mitochondria are representative of the individual cells. Therefore, in order to increase the coverage of various cells, we considered a maximum of two mitochondria from each cell which were randomly chosen. This part is modified in the methods section '**Quantification of various parameters related to cristae morphology**' to make it clear. Thus, while the quantification of various parameters including dynamics involved individual mitochondria, various cells were classified as belonging to normal or enlarged category while measuring ATP levels.

5. What is the average size of all mitochondria per cell? Is this addressed in Figure 2B or are only analyzed mitochondria included? Please clarify. Were the mitochondria chosen for analysis representative for the respective cell?

Response: The data obtained by super-resolution imaging of mitochondria is used for quantifying cristae dynamics which is a very challenging and time-consuming method done in a blind-manner. As mentioned in **response 4B**, the cristae morphology is fairly uniform in individual cells, therefore, we only included the mitochondria which were analysed for various cristae parameters in our analysis which are really huge data-sets already. Thus, the average

size of individual mitochondria per cell are not represented while analysing images obtained with STED SR imaging. Please also check response 4B.

<minor comments>

1. explain the mt-Go-AT team2, what is GFP (green fluorescent protein) and OTP (?)

Response: GFP is Green Fluorescent Protein and OFP is Orange Fluorescent protein and included in the revised text.

2. the graphs show in principle, e.g. Fig.1B, 3B-E show events/mitochondrion as far as I understand, not per cristae.

Response: Thank you for pointing this out. It is actually the average number of events per cristae per mitochondria. We have changed the Y-axis to events/cristae/mito in **Fig 1C** (previous 1B), **Fig 3B-E** and wherever applicable for other figures throughout the manuscript.

Figure 1C

Figure 3B-E

3. I would recommend changing the legend of the x-axis of Fig.2B-F to mito-width (y-axis could be probability density function, PDF).

Response: We have now changed the X-Axis to mito width (originally width) in **Fig 2B-F**. The Y-axis are still retained as percentage mitochondria where cells treated with few mitochondrial toxins do not show a gaussian distribution of mitochondrial width.

Figure 2B-F

****Referees cross-commenting****

both expert opinions address similar concerns and therefore a revision should be requested

Reviewer #1 (Significance (Required)):

The study is thorough and the experiments and results are well described. Overall, however, it remains a descriptive study and does not provide mechanisms. There is also no discussion of how MMP-dependent proteins, such as Opa1, which was previously studied by the Reichert group, might be affected. For swelling mechanisms, the opening of the mitochondrial permeability transition pore was discussed. This could be tested using inhibitors, but perhaps not within the scope of this publication. Nevertheless, the information provided by the study is of interest to the bioenergetics community and should be made available.

Response: Thank you for the overall inputs.

We tested the processing of OPA1 forms and found that after 30 mins, only CCCP treatment led to the processing of long isoforms to short forms (**Fig S6C**). We now included in the discussion that it is possible that short OPA1-forms are correlative to increased cristae merging as well as splitting events upon treatment with CCCP.

New Figure S6C

Reviewer #2 (Evidence, reproducibility and clarity (Required)):

Summary:

The authors investigated cristae merging and splitting events using ultra-resolution STED. The goal was to test if cristae membrane remodeling is dependent on OXPHOS complexes, mitochondrial membrane potential ($\Delta\Psi_m$), and the ADP/ATP nucleotide translocator. To do this the authors utilized several mitochondrial toxins with known mechanisms of action. Interestingly, many changed overall cristae density but did not change the cristae remodeling events. Inhibition of ANT did change cristae morphology and cristae dynamics.

Major Concerns

1. Many conclusions and concepts need more clarification. For example, a major take home from the abstract is that various ETC inhibitors and protonophores reduce cristae density but not did not change cristae remodeling events. If cristae density is reduced, how can this occur without cristae remodeling events? Remodeling events need to be clearly defined in the introduction and abstract.

Response: Thank you for pointing out this lack of sharpness in our terminology which indeed can cause a misunderstanding. To avoid this, we have now included 'changes in cristae morphology' as well as 'dynamic merging and splitting events of cristae' under the broader term cristae remodelling. Thus, we had changed the wording 'cristae remodeling' to cristae dynamics in the abstract and wherever appropriate in the manuscript text.

The cristae morphology analysis showed no change in cristae area (**Fig S3C**) which was accompanied by mitochondrial enlargement. Therefore, cristae density was reduced. For the purpose of clarity, we added a sentence in the introduction section while giving a peek into our results that 'cristae dynamic events are ongoing despite reduced cristae density'. In addition, we have now included in the results section the following statement: 'Cristae membrane remodeling has been used to describe cristae dynamic events (i.e. cristae merging and splitting) as well as overall changes in cristae morphology within a single mitochondrion in this manuscript'.

Figure S3C and D

2. Other interpretations are also unclear such as how ETC inhibitors which reduce ATP levels did not impact cristate remodeling events, yet inhibiting ATP/ADP exchange did greatly impact this phenomenon. It seems likely that the inhibition of ANT has nothing to do with ATP/ADP exchange since most of the ETC inhibitors no doubt greatly impact overall ATP/ADP exchange. This interpretation needs clarification.

Response: We agree that further clarification is needed, in particular to explain why ATP/ADP exchange is actually ongoing even when OXPHOS inhibitors are applied and to explain why reduced ATP levels do not mean that there is no ATP/ADP exchange occurring. Treatment of HeLa cells with various mitochondrial toxins inhibiting the function of OXPHOS complexes leads to decreased ATP levels due to ongoing ATP consumption within the cell (**Fig 4**). One should also consider that two things can and do happen when most of these toxins are applied regarding ATP exchange. First, the ATPase can act in reverse mode which is a (partial) compensatory mechanism to restore $\Delta\Psi_m$ and which will further decrease ATP levels (Note: not in the presence of oligomycin). Second, under these conditions ADP/ATP exchange is still ongoing in order to transport ATP derived from glycolysis in the cytosol to the mitochondrial matrix which also causes an (partial) compensatory increase in membrane potential. After ATP import ATP is hydrolysed to ADP for reverse proton pumping via the F1FO-ATPase or alternatively by the F1-part alone without proton pumping. In all these cases it is essential and possible to exchange ADP with ATP constantly. Therefore, the overall exchange of ADP and ATP is not necessarily grossly expected to be different when compared to untreated cells (due to compensatory glycolysis and subsequent ATP import and hydrolysis in the matrix). On the other hand, BKA treatment which clearly impairs the exchange of ADP and ATP will lead to a completely different situation compared to only treating with OXPHOS inhibitors. With BKA the mitochondrial matrix cannot anymore be resupplemented with ATP derived from glycolysis and metabolite flux is grossly hampered. Consistent with this a strong reduction in $\Delta\Psi_m$ and oxygen consumption is accompanied with BKA treatment (Fig. 5AB & SFig 7F). Thus, w.r.t cristae dynamic events, in the time-frame we used for imaging, a reduction of ATP levels does not impede occurrence of cristae merging and splitting events while BKA treatment does (**Fig S7**). We discuss this indeed interesting and unexpected finding in the discussion section. We propose that rather ongoing metabolite flux (ATP/ADP exchange) is critical for maintaining cristae dynamics and blocking it is detrimental for it. We adapted the discussion in this direction to make it more clear.

D

Figure S7A, B and D

3. Why did the authors wait 30 min to image after the addition of mitochondrial toxins? I would have guessed there is a more rapid change in response to these inhibitors. Is there a chance he authors missed the most dramatic events?

Response: Since we were inclined to observe early responses, cells were imaged within the first 30 mins after addition of the respective mitochondrial toxins (Please **see methods 'cell culture transfection and mitochondrial toxin treatment'**). Thus, to answer this question we want to emphasize that we did not wait 30 minutes but we restricted our time frame of analysis to 30 min. Therefore, we think that we did not miss out on any rapid changes occurring early on. Regarding this point, Reviewer #1 (Query 3) asked for responses at a later time-point. Please read the Reviewer #1, response 3B.

4. How do these mitochondrial toxins that are known to cause mitochondrial swelling not induce changes in cristate density?

Response: Thank you for the question. Probably, there is a misunderstanding. In **Fig S3E**, we clearly show that as the mitochondrial width increases in cells after treatment with mitochondrial toxins, there is a clear decrease in cristae density. In fact, the reduced cristae density is observed exclusively in enlarged mitochondria.

Figure S3E-I

5. It's interesting that inhibition of the ANT translocator by BKA treatment led to increased percentage of mitochondria with abnormal cristae morphology. It's accepted that inhibition of

ANT profoundly reduces mitochondrial swelling. Do the authors have any data suggesting that abnormal cristae morphology actually is a mechanism for reducing cell death events such as permeability transition? Did the authors utilize cyclosporin A concomitantly with any of the mitochondrial toxins?

Response: This is a very interesting question! As the reviewer might be aware, there is evidence connecting cristae remodelling to induction of apoptosis (3). Cristae transitioned to a highly interconnected state after tBID treatment within minutes. However, it is unclear what is the contribution of cristae dynamics in this regard. Within 30 mins, there were no visual signs of cell death in our experiments as observed under a microscope. Hence, we did not use cyclosporin A in our experiments. In our opinion, this question will form part of a very interesting future study and is currently beyond the scope of this manuscript.

6. Are the authors confident in the data given many of the experiments utilized quantification of 10-20 mitochondria? How are you sure this sampling is sufficient for phenomenon being studied?

Response: Please see Reviewer 1, Response 4B. As pointed in the response to reviewer #1, the cristae morphology is fairly uniform in individual cells. Therefore, in order to maximise the cell population covered, we randomly used a maximum of two mitochondria from each cell. In addition, we included cristae analysis from at least three biological replicates in order to observe the reproducibility of the data. Taking these factors into consideration, we are confident that our results reflect a sufficient sample size. Further, we would like to point out while our group performs STED super-resolution imaging routinely, the quantification of cristae merging and splitting events done in a blind yet manual manner is a really laborious and time-consuming process. In the future, we are also looking to optimise this at least in a semi-automated manner.

7. Figure 4 and 5 merely confirm current dogma and don't really contribute to the overall conclusions and can be moved to supplemental data.

Response: We agree that **Fig 5** is confirming to the current dogma. Therefore, we moved it to **Fig S6**. Regarding Fig 4, we would like to highlight that there is a decrease of ATP levels before mitochondria enlarge. Thus, we would like to retain it as part of the main figure.

8. It's interesting that BKA dose dependently decreased ATP-linked respiration and all doses limited maximal respiratory capacity. It would be interesting to know if the BKA normal vs. abnormal mitochondria have differential membrane potential?

Response: Thank you for the interesting question. Overall, BKA treatment leads to a significant decrease of $\Delta\Psi_m$ in the whole cell population (**Fig S7**). Further, the abnormal cristae morphology is only seen in one-third of the population of mitochondria (Fig shown in Response 2). Thus, a drop in $\Delta\Psi_m$ seems to be a very early response upon exposure to BKA and independent of cristae morphology. An ideal experiment to address this question would be to image cristae dynamics and $\Delta\Psi_m$ using super-resolution imaging which is challenging according to the state-of-art and available chemicals.

Figure S7E and F

9. Overall, this is an interesting study and seems appropriately performed but the results and conclusions are unclear. More discussion should include physiological relevance and impact and how this data influences previous work. Some physiological perturbations beyond the mitochondrial toxins and or utilization of genetic models would strengthen the interpretation and overall impact.

Response: Thank you. We added an OPA1 blot showing the different L-OPA1 and S-OPA1. (Reviewer #1, response in significance section) where we observed that S-OPA1 cleavage is selectively enhanced in CCCP-treated cells which could be correlated with enhanced cristae dynamics. We also included these results in the main text.

New Figure S6C

****Referees cross-commenting****

Yes, I conclude that given the significant overlap in reviewer comments and general need for

clarification of concepts and data that a revision is in order.

Reviewer #2 (Significance (Required)):

Overall, a highly specialized study with audience limited to mitochondriacs. Although, I'll note tis is a hot area of study and there is high interest in the field. Some of the data interpretation is difficult to understand and overall more context is needed to explain the results, impact and relevance. Defining exactly what a cristae remodeling event is and how this differs from cristae density and how the two aren't directly connected is unclear.

Review by a mitochondrial biologist specializing in mitochondrial signaling and connection to physiology.

1. Baker MJ, Lampe PA, Stojanovski D, Korwitz A, Anand R, et al. 2014. Stress-induced OMA1 activation and autocatalytic turnover regulate OPA1-dependent mitochondrial dynamics. *EMBO J* 33: 578-93
2. Farkas DL, Wei MD, Febroriello P, Carson JH, Loew LM. 1989. Simultaneous imaging of cell and mitochondrial membrane potentials. *Biophys J* 56: 1053-69
3. Scorrano L, Ashiya M, Buttle K, Weiler S, Oakes SA, et al. 2002. A distinct pathway remodels mitochondrial cristae and mobilizes cytochrome c during apoptosis. *Dev Cell* 2: 55-67

October 11, 2023

Re: Life Science Alliance manuscript #LSA-2023-02386

Dr. Arun Kumar Kondadi
Heinrich Heine University Duesseldorf;
Universitätsstrasse 1
Düsseldorf, North Rhine Westphalia 40225
Germany

Dear Dr. Kondadi,

Thank you for submitting your revised manuscript entitled "Live-cell super-resolution nanoscopy reveals modulation of cristae dynamics in bioenergetically compromised mitochondria" to Life Science Alliance. The manuscript has been seen by one of the original reviewers whose comments are appended below. While the reviewer continues to be overall positive about the work in terms of its suitability for Life Science Alliance, some important issues remain.

Our general policy is that papers are considered through only one revision cycle; however, given that the suggested changes are relatively minor, we are open to one additional short round of revision. Please note that I will expect to make a final decision without additional reviewer input upon re-submission.

Please submit the final revision within one month, along with a letter that includes a point by point response to the remaining reviewer comments.

To upload the revised version of your manuscript, please log in to your account: <https://lsa.msubmit.net/cgi-bin/main.plex>
You will be guided to complete the submission of your revised manuscript and to fill in all necessary information.

B. MANUSCRIPT ORGANIZATION AND FORMATTING:

Sincerely,

Reviewer #1 (Comments to the Authors (Required)):

The authors have done a good job in responding to the queries and have added valuable new data. However, the interpretation

of observations and integration with existing knowledge needs to be improved. This is especially true in the discussion section. The conclusion: "Decreased cristae density along with no change in cristae area in enlarged mitochondria indicates presence of longer cristae compared to normal mitochondria in all treatment conditions" is a bit misleading, rather enlarged mitochondria with the same total area of cristae means first of a reduced cristae density per mitochondrial area, which is cristae depletion. This is shown in FigS3? What indication do the authors have that cristae became longer? Opa1 loss leads indeed to cristae depletion and less cristae junctions according to previous work, which should be cited (Kushnareva et al., 2013)

The authors describe that loss of MMP and reduced OCR do not appear to affect cristae dynamics, but CCCP treatment increases cristae dynamics (p13). As a possible explanation, they suggest that increased electron flow through ETC complexes SOMEHOW stimulates cristae dynamics? Is there a molecular mechanism behind this, or is cristae dynamics a purely stochastic and temperature-dependent process? ...please comment

Next, the authors ask about the functional interplay between ATP levels and cristae dynamics (p. 13, line 418). Their comment is "probably increased oxygen consumption and cristae dynamics regulate ATP levels or vice versa". This does not really clarify the question.

Next, they propose that cristae dynamics may be determined by structural constraints and suggest that densely packed cristae would impose a constraint that limits cristae dynamics. Since they find a decrease in cristae density in all conditions (Fig. S3), why aren't cristae dynamics increased in all conditions, but only in CCCP-treated mitochondria? One difference could be the OPA processing in CCCP-treated mitochondria, which could allow the loosening of the OPA1 scaffold in cristae junctions (Faelber et al., 2019). If OMA1 and YME1L peptidases are ATP-dependent and ATP is higher in CCCP-treated cells, this would be a molecular link. Please do some literature research.

In sum, albeit this is a thorough study, the interpretation of the data is still rather preliminary.

Faelber, K., Dietrich, L., Noel, J. K., Wollweber, F., Pfitzner, A. K., Muhleip, A., . . . Daumke, O. (2019). Structure and assembly of the mitochondrial membrane remodelling GTPase Mgm1. *Nature*, 571(7765), 429-433. doi:10.1038/s41586-019-1372-3
Kushnareva, Y. E., Gerencser, A. A., Bossy, B., Ju, W. K., White, A. D., Waggoner, J., . . . Bossy-Wetzler, E. (2013). Loss of OPA1 disturbs cellular calcium homeostasis and sensitizes for excitotoxicity. *Cell Death Differ*, 20(2), 353-365. doi:10.1038/cdd.2012.128

The authors have done a good job in responding to the queries and have added valuable new data.

Response: Thank you!

- 1) However, the interpretation of observations and integration with existing knowledge needs to be improved. This is especially true in the discussion section.

The conclusion: "Decreased cristae density along with no change in cristae area in enlarged mitochondria indicates presence of longer cristae compared to normal mitochondria in all treatment conditions" is a bit misleading, rather enlarged mitochondria with the same total area of cristae means first of a reduced cristae density per mitochondrial area, which is cristae depletion. This is shown in FigS3? What indication do the authors have that cristae became longer?

Response: We thank the reviewer for this valuable comment. In retrospect, the reviewer is right as the mentioned discussion should be taken in the context of enlarged mitochondria. We have now changed the statement accordingly in the discussion. 'Therefore, cristae density was reduced due to an overall increase in mitochondrial area but not due to changes in cristae area.' As we did not quantify the length of cristae, we removed this statement.

- 2) Opa1 loss leads indeed to cristae depletion and less cristae junctions according to previous work, which should be cited (Kushnareva et al., 2013)

Response: It is informative to add that not only the ratio of S-OPA1 and L-OPA1 but also absolute OPA1 levels are related to cristae morphology and dynamics. Accordingly, we have added the following statement in the discussion: 'At the level of cristae morphology, it is known that depletion of OPA1 leads to reduced number of cristae and CJs (Kushnareva YE et al, 2013) and disorganized cristae (Olichon A et al, 2003). Accordingly, it has been shown that cristae dynamics are reduced in OPA1 KO cells (Hu C et al, 2020)'.

- 3) The authors describe that loss of MMP and reduced OCR do not appear to affect cristae dynamics, but CCCP treatment increases cristae dynamics (p13). As a possible explanation, they suggest that increased electron flow through ETC complexes SOMEHOW stimulates cristae dynamics? Is there a molecular mechanism behind this, or is cristae dynamics a purely stochastic and temperature-dependent process?please comment

Response: We speculated that the increased electron flow upon CCCP treatment might be one criterion which increased cristae dynamics in the manuscript version before. We agree that we do not know how this could be mediated mechanistically but feel it is important to put forward this possibility. We do discuss now in more detail a molecular mechanism involving OPA1 cleavage which was proposed in the initial part of the same paragraph but was probably not sufficiently clear. Therefore, we modified this part by re-introducing OPA1 cleavage as a possible mechanism briefly at the end of the paragraph. Thus, we do not think that the cristae dynamics is a stochastic phenomenon. Overall, we wanted to point out that cristae dynamics is better correlated to metabolite flux (of e.g. ADP) compared to the pure enzymatic activity of ETC complexes.

The modified para 'Another criterion which has already been introduced is that the cristae dynamics might be regulated by OPA1 cleavage which was only observed in CCCP treatment and not in treatments with other mitochondrial toxins within 30 min. Thus, OPA1 cleavage could be a possible mechanism for regulating cristae dynamics'.

- 4) Next, the authors ask about the functional interplay between ATP levels and cristae dynamics (p. 13, line 418). Their comment is "probably increased oxygen consumption and cristae dynamics regulate ATP levels or vice versa". This does not really clarify the question.

Response: We thank the reviewer for this concern and agree that this sentence does not clarify this question. Therefore, we have removed this sentence. While we clearly show the connection between cristae dynamics with the cellular metabolic status, more studies need to be done to understand the spatio-temporal regulation of ATP levels and cristae dynamics. In this section we only want to emphasize that under conditions of high oxygen consumption (and maintained ATP levels) cristae dynamics is increased. This is to prepare (for later) when we discuss below, how rather ADP/ATP metabolic flux - as opposed to ATP levels - could be a determinant of cristae dynamics. This section has been modified as well and we hope that it is clearer now.

5) Next, they propose that cristae dynamics may be determined by structural constraints and suggest that densely packed cristae would impose a constraint that limits cristae dynamics. Since they find a decrease in cristae density in all conditions (Fig. S3), why aren't cristae dynamics increased in all conditions, but only in CCCP-treated mitochondria? One difference could be the OPA processing in CCCP-treated mitochondria, which could allow the loosening of the OPA1 scaffold in cristae junctions (Faelber et al., 2019). If OMA1 and YME1L peptidases are ATP-dependent and ATP is higher in CCCP-treated cells, this would be a molecular link. Please do some literature research.

Response: We agree with the reviewer and clearly explain that structural constraints cannot be the only determinant. We hope that with the modifications in the discussion section, this section is better explained. This question is also related to the comment #3. We have proposed in discussion the following: 'Overall, mitochondrial enlargement was **necessary but not sufficient** to display enhanced cristae membrane dynamics and these data point to the possibility that conditions of high oxygen consumption, which is equivalent to high electron flow from NADH to oxygen in the respiratory chain, may be one criterion to promote cristae merging and splitting events'. We further added, in this round of revision, in the discussion reiterating that 'the cristae dynamics might be regulated by OPA1 cleavage which was only observed in CCCP treatment and not in treatments with other mitochondrial toxins within 30 min. Thus, OPA1 cleavage could be a possible mechanism for regulating cristae dynamics'.

We also incorporated the suggestion in the manuscript suggested by the reviewer in the discussion. 'It has been shown that a balance of L-OPA1 and S-OPA1 keep CJs tight (Frezza C et al, 2006). Furthermore, it was demonstrated that S-Mgm1 (homolog of human OPA1) has the ability to form helical lattice both on the inside as well as outside of lipid tubes (Faelber K et al, 2019). In addition, it could be either a left- or right-handed helix. Both these properties contribute to exert a constricting as well as pulling force which were proposed to play important roles not only in inner membrane fusion and fission but also in cristae stabilization. At the level of cristae morphology, it is known that depletion of OPA1 leads to reduced number of cristae and CJs (Kushnareva YE et al, 2013) and disorganized cristae (Olichon A et al, 2003). Accordingly, it has been shown that cristae dynamics are reduced in OPA1 KO cells (Hu C et al, 2020)'.

6) In sum, albeit this is a thorough study, the interpretation of the data is still rather preliminary.

Response: We think that we have now considerably improved in this respect, with the help of the reviewer, leading to our interpretations being valid and clear. We are convinced that the models we propose here provide a useful basis for future research in this emerging field.

Faelber, K., Dietrich, L., Noel, J. K., Wollweber, F., Pfitzner, A. K., Muhleip, A., . . . Daumke, O. (2019). Structure and assembly of the mitochondrial membrane remodelling GTPase Mgm1. *Nature*, 571(7765), 429-433. doi:10.1038/s41586-019-1372-3

Kushnareva, Y. E., Gerencser, A. A., Bossy, B., Ju, W. K., White, A. D., Waggoner, J., . . . Bossy-Wetzel, E. (2013). Loss of OPA1 disturbs cellular calcium homeostasis and sensitizes for excitotoxicity. *Cell Death Differ*, 20(2), 353-365. doi:10.1038/cdd.2012.128

References:

Faelber K, Dietrich L, Noel JK, Wollweber F, Pfitzner AK, Muhleip A, Sanchez R, Kudryashev M, Chiaruttini N, Lilie H, et al (2019) Structure and assembly of the mitochondrial membrane remodelling gtpase mgm1. *Nature* 571: 429-433. doi:10.1038/s41586-019-1372-3

Frezza C, Cipolat S, Martins de Brito O, Micaroni M, Beznoussenko GV, Rudka T, Bartoli D, Polishuck RS, Danial NN, De Strooper B, et al (2006) Opa1 controls apoptotic cristae remodeling independently from mitochondrial fusion. *Cell* 126: 177-189. doi:10.1016/j.cell.2006.06.025

Hu C, Shu L, Huang X, Yu J, Li L, Gong L, Yang M, Wu Z, Gao Z, Zhao Y, et al (2020) Opa1 and micos regulate mitochondrial crista dynamics and formation. *Cell death & disease* 11: 940. doi:10.1038/s41419-020-03152-y

Kushnareva YE, Gerencser AA, Bossy B, Ju WK, White AD, Waggoner J, Ellisman MH, Perkins G, Bossy-Wetzel E (2013) Loss of opa1 disturbs cellular calcium homeostasis and sensitizes for excitotoxicity. *Cell death and differentiation* 20: 353-365. doi:10.1038/cdd.2012.128

Olichon A, Baricault L, Gas N, Guillou E, Valette A, Belenguer P, Lenaers G (2003) Loss of opa1 perturbs the mitochondrial inner membrane structure and integrity, leading to cytochrome c release and apoptosis. *The Journal of biological chemistry* 278: 7743-7746. doi:10.1074/jbc.C200677200

November 2, 2023

RE: Life Science Alliance Manuscript #LSA-2023-02386R

Dr. Arun Kumar Kondadi
Heinrich Heine University Düsseldorf
Universitätstrasse 1
Düsseldorf, North Rhine Westphalia 40225
Germany

Dear Dr. Kondadi,

Thank you for submitting your revised manuscript entitled "Cristae dynamics are modulated in bioenergetically compromised mitochondria". We would be happy to publish your paper in Life Science Alliance pending final revisions necessary to meet our formatting guidelines.

- please add the Twitter handle of your host institute/organization as well as your own or/and one of the authors in our system
- please upload a manuscript file without tracking changes
- there is a call-out for figure 7A, B but there is no figure 7 -- please correct
- please add a call-out for Figure 6 to your main manuscript text

Figure Checks:

- Figure S2 is the same as Figure 1A and Figure S4A is the same as Figure 3A. Why is this data needed in 2 separate figures?

A. FINAL FILES:

B. MANUSCRIPT ORGANIZATION AND FORMATTING:

Sincerely,

November 6, 2023

RE: Life Science Alliance Manuscript #LSA-2023-02386RR

Dr. Arun Kumar Kondadi
Heinrich Heine University Düsseldorf
Universitätstrasse 1
Düsseldorf, North Rhine Westphalia 40225
Germany

Dear Dr. Kondadi,

Thank you for submitting your Research Article entitled "Cristae dynamics are modulated in bioenergetically compromised mitochondria". It is a pleasure to let you know that your manuscript is now accepted for publication in Life Science Alliance. Congratulations on this interesting work.

DISTRIBUTION OF MATERIALS:

Again, congratulations on a very nice paper. I hope you found the review process to be constructive and are pleased with how the manuscript was handled editorially. We look forward to future exciting submissions from your lab.

Sincerely,
